# Vision-Language Dataset Distillation

**Xindi Wu**[1]    **Byron Zhang**[1]    **Zhiwei Deng**[2]    **Olga Russakovsky**[1]

[1]*Princeton University*    [2]*Google DeepMind*

*{xindiw, zishuoz, olgarus}@princeton.edu, zhiweideng@google.com*

**Reviewed on OpenReview:** *https://openreview.net/forum?id=6L6cD65pot*

**Website:** *https://princetonvisualai.github.io/multimodal_dataset_distillation/*

## Abstract

Dataset distillation methods reduce large-scale datasets to smaller sets of synthetic data, preserving sufficient information to quickly train a new model from scratch. However, prior work on dataset distillation has focused exclusively on image classification datasets, whereas modern large-scale datasets are primarily vision-language datasets. In this work, we design the first vision-language dataset distillation method, building on the idea of trajectory matching. A key challenge is that vision-language datasets do not have a set of discrete classes. To overcome this, our proposed method jointly distills image-text pairs in a contrastive formulation. Further, we leverage Low-Rank Adaptation (LoRA) matching to enable more efficient and effective trajectory matching in complex modern vision-language models. Since there are no existing baselines, we compare our distillation approach with three adapted vision-language coreset selection methods. We demonstrate significant improvements on the challenging Flickr30K and COCO retrieval benchmarks: for example, on Flickr30K, the best coreset selection method selecting 1000 image-text pairs for training achieves only 5.6% image-to-text retrieval accuracy (i.e., recall@1); in contrast, our dataset distillation almost doubles that to 9.9% with just 100 training pairs, an order of magnitude fewer.

## 1 Introduction

$$Data = Information + Irrelevant\ Data \qquad (\text{Wright \& Ma}, 2022)$$

Dataset distillation aims to create concise summaries of data that preserve most of the critical information of the entire dataset. It holds paramount importance in the era of big data as it addresses the challenge posed by "*Data = Information + Irrelevant Data*" (Wright & Ma, 2022), where we often need to learn the useful information in an ocean of non-critical data. The recent growth of dataset distillation methods, e.g., (Wang et al., 2018; Cazenavette et al., 2022; Nguyen et al., 2020) has primarily focused on image classification datasets, capturing class-specific information to build discriminative boundaries. Considering the recent progress in multimodal machine learning, where we witness the explosion of vision-language datasets in which the majority of image pixels may belong to irrelevant contextual elements and may further lack corresponding textual descriptions, a significant necessity arises to efficiently distill this vast amount of data. A well-distilled multimodal dataset simplifies complex vision-language interactions and emphasizes the most salient connections, making it more effective for models to learn cross-modal representations.

**Why is it hard?** The first key challenge and the main difference from prior dataset distillation methods (Wang et al., 2018; Cazenavette et al., 2022) is that vision-language datasets do not contain a discrete set of classes to ground the distillation process. Instead, these datasets contain complex cross-modal connections and redundancies, requiring a co-distillation approach to capture their interdependencies effectively. Second, the complexity of cross-modal representations and vision-language models (VLMs) leads to computational challenges. Prior dataset distillation methods operate on low-resolution images (typically 28x28 or 32x32, as in MNIST (LeCun et al., 1998) or CIFAR (Krizhevsky et al., 2009)) and nevertheless suffer from significant computational costs, even with ConvNet when creating distilled datasets. Vision-language datasets often

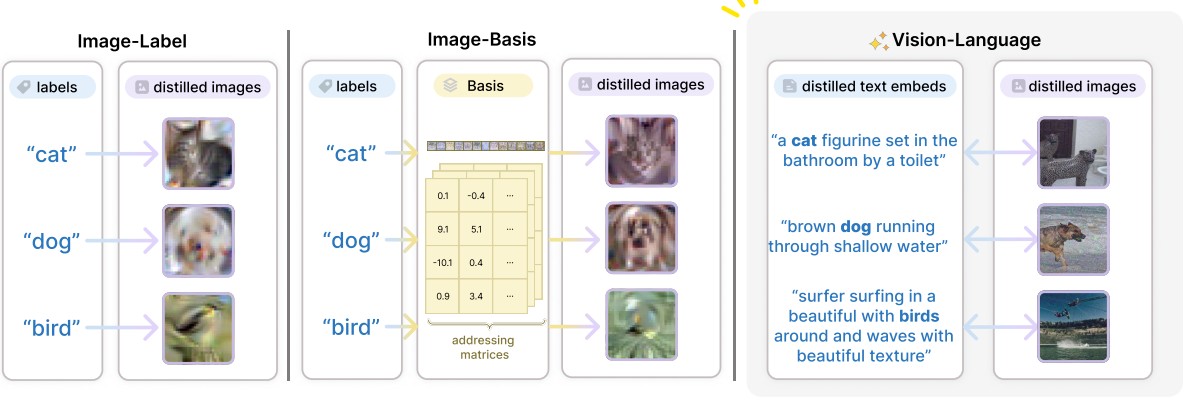

Figure 1: **Dataset Distillation Comparison.** (*Left*) Prior dataset distillation methods (Wang et al., 2018; Cazenavette et al., 2022; Nguyen et al., 2020) are class-specific: they distill the key information for each individual discrete class. (*Center*) Even the recently developed method (Deng & Russakovsky, 2022), which enables information sharing between classes through learned bases, still assumes a discrete set of classes. (*Right*) In contrast, we set out to distill vision-language datasets with no discrete classes; we do so via a novel method which jointly distills images and texts.

contain higher-resolution images, and models designed for vision-language tasks are substantially more complex, such as Vision Transformers (ViTs) (Dosovitskiy et al., 2020). Lastly, unlike continuous data, text is inherently non-differentiable, making direct gradient-based optimization impossible on discrete text tokens.

**Our work.** We propose the first Vision-Language Dataset Distillation method. Concretely, given a dataset of images with corresponding text descriptions, our method creates a much smaller synthetic set of (image, text embedding) pairs which can then be used to efficiently train a model that aims to learn the image-text alignment. Given the infeasibility of direct information extraction, our co-distillation is achieved by implicitly matching the by-products of the target vision-language data and the synthetic ones. In our case, the by-product is the *long-range training bi-trajectory*. Additionally, as finetuning pretrained models is widely used in vision-language tasks, we match the trajectories of the low-rank matrices (Hu et al., 2021) for complex models to effectively distill critical information.

**Contributions.** To the best of our knowledge, this is the first work to tackle vision-language dataset distillation. In doing so, we make the following key contributions:

1. We highlight the challenges of vision-language dataset distillation and establish the first set of baselines for this task by adapting three coreset selection methods (Welling, 2009; Toneva et al., 2018; Farahani & Hekmatfar, 2009; Sener & Savarese, 2017).
2. We propose the Bi-Trajectory Vision-Language Co-Distillation method. Different from prior image classification dataset distillation methods, our method is not restricted to discrete classes and distills vision-language pairs jointly. We leverage Low-Rank Adaptation (LoRA) matching to make it computationally feasible for training with complex models (e.g., ViTs) on high-resolution images.
3. Our method significantly improves image-text retrieval with training set constraints on the challenging Flickr30K (Plummer et al., 2015) and COCO (Lin et al., 2014) datasets. For example, the best coreset selection method (adapted K-center) achieves 5.6% image-to-text retrieval performance (R@1) after selecting 1000 image-text pairs for training. In contrast, our method almost doubles that performance on the same task to 9.9% with **an order of magnitude** fewer (just 100) distilled image-text pairs.

The growing interest in multimodal datasets makes it even more crucial to develop mechanisms that efficiently and effectively distill insights from different modalities. We hope this work jump-starts further research into the important and challenging space of vision-language dataset distillation.

## 2   Related Works

**Dataset Distillation.** The concept of dataset distillation has demonstrated that a handful of synthetic images, although not drawn from the training distribution, can achieve comparable performance to that of the original dataset (Wang et al., 2018). Meta-learning based data distillation approaches (Nguyen et al., 2021; Zhou et al., 2022; Deng & Russakovsky, 2022; Nguyen et al., 2020; Vicol et al., 2022; Zhou et al., 2022) typically use bilevel optimization, where the inner loop trains on the distilled data samples and the outer loop optimizes meta datasets. Several works (Zhao & Bilen, 2021b;a; Cazenavette et al., 2022; Jiang et al., 2022; Du et al., 2023; Liu et al., 2023) explored by-product matching approaches, such as matching the gradient or trajectory of the gradient with respect to the model trained on the real and distilled data.

Our work is mostly inspired by the trajectory matching method (Cazenavette et al., 2022; Cui et al., 2022), which is more efficient for optimization since they mostly do not involve long unrolling of computation graphs. Rather than aligning model gradients, another thread of work (Zhao & Bilen, 2021b; Wang et al., 2022; Lee et al., 2022) has been developed to align feature distributions between real and distilled data using a distribution divergence metric in the latent space. While most of the prior works focus on image classification dataset distillations, (Sucholutsky & Schonlau, 2021) explored dataset distillation on text datasets. Our work is the first to scale up dataset distillation methods to vision-language datasets, which involves creating distilled data that capture critical features and complex relationships within and between two modalities.

**Cross-modal Retrieval.** Most cross-modal retrieval methods function at the representation level and encourage a joint embedding space by measuring the similarities between learned representations across different modalities (Liang et al., 2022; Zhu et al., 2022; Pokle et al., 2022; Chun et al., 2021; Wu et al., 2023). Image-text retrieval focuses on retrieving images given captions, or of captions given images (Wang et al., 2020b; Wu et al., 2019). Many techniques have been developed to produce representations that are semantically similar for image-text pairs (Huang et al., 2018; Gu et al., 2018). More advanced image-text alignment methods (Li et al., 2022; Lin et al., 2023; Pandey et al., 2022) that incorporate pretraining have shown promising results on image-text retrieval tasks. We evaluate our vision-language dataset distillation method on image-text retrieval tasks.

**Vision-language Knowledge Distillation.** Prior efforts on vision-language distillation are primarily centered around knowledge distillation, which transfers knowledge from a larger teacher model to a smaller student model to improve the latter's performance (Xue et al., 2023; Radenovic et al., 2023; Valverde et al., 2021). Our dataset distillation study focuses on the orthogonal question and is fundamentally a pragmatic compression problem. We aim to find equivalent bits that can represent the entire vision-language datasets.

## 3   Method

We propose a vision-language dataset distillation method for distilling a large-scale dataset consisting of (image, text) pairs into a smaller dataset, while maintaining much of the original dataset's information relevant to training vision-language models (VLMs). The detailed method is in Fig. 2.

### 3.1   Problem Formulation

Consider a large-scale dataset $\mathbf{D} = \{(x_i, y_i)\}_{i=1}^{N}$, where each $x_i$ denotes an image and each $y_i$ denotes its corresponding text descriptions; note that in practice, $y_i$ may be a set $\{y_{i1}, y_{i2}, ..., y_{iK}\}$ where $K$ is the number of descriptions associated with each image. Our goal is to learn a smaller dataset $\hat{\mathbf{D}} = \{(\hat{x}_j, \hat{y}_j)\}_{j=1}^{M}$, with significantly fewer data pairs $M \ll N$ that still captures most of the essential information needed to train a VLM effectively. For $\hat{y}_i$, we aim to use one (instead of $K$) sentence per image in the distilled set for a more compact representation. Concretely, consider a VLM with vision encoder $f(\cdot; \theta_{img})$ and language encoder $g(\cdot; \theta_{txt})$. This model can be trained by optimizing the similarity loss which encourages alignment between the image and text embeddings:

$$\theta^* \approx \arg\min_{\theta} \frac{1}{N} \sum_{i=1}^{N} \ell\left(f(x_i; \theta_{img}), g(y_i; \theta_{txt})\right). \tag{1}$$

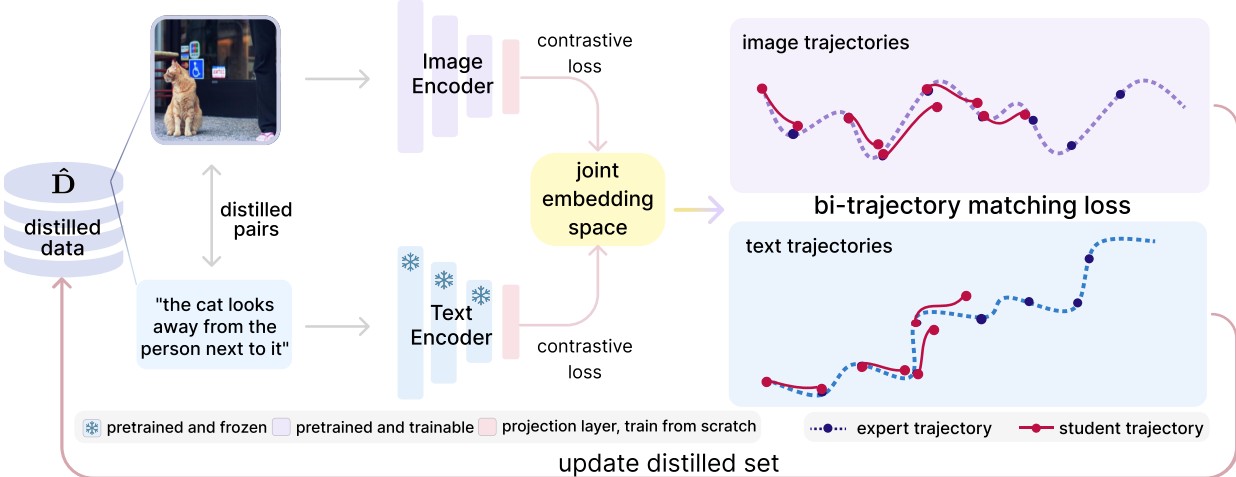

Figure 2: **Vision-Language Dataset Distillation.** Both the image and text encoders are pretrained and followed by a trainable projection layer, and the text encoder is frozen. We use contrastive loss to measure the distance between the paired image-text embeddings, which influences the trajectory updates during distillation. The right panel shows how the distilled data aligns its training trajectory with the expert's, from a random starting point on the expert trajectory. The distilled dataset is updated based on bi-trajectory matching loss between the student and expert parameter trajectories.

Our goal is to distill a dataset $\hat{\mathbf{D}}$ such that the model trained with $\hat{\mathbf{D}}$ obtains comparable vision-language matching performance as the one trained on $\mathbf{D}$. More specifically, consider a metric $\mathbf{m}$ defined to quantify the correlation between the model's representation $f(x; \theta_{img})$ of a given image $x$ and the representation $g(y; \theta_{img})$ of a given text $y$, this representation should match the actual similarity between the image and text pairs. The correlation calculation is based on whether the image-text pair is a positive (matching) or a negative (non-matching) pair. Given the test dataset $\mathbf{D}_{test}$, our objective can be defined as follows:

$$\mathbb{E}_{(x,y)\sim\mathbf{D}_{test}}\big[\mathbf{m}(f(x;\theta^*_{img}), g(y;\theta^*_{txt}))\big] \simeq \mathbb{E}_{(x,y)\sim\mathbf{D}_{test}}\big[\mathbf{m}(f(x;\hat{\theta}_{img}), g(y;\hat{\theta}_{txt}))\big], \quad (2)$$

where $\theta^*$ represents the optimal model parameters from training on the entire dataset, and $\hat{\theta}$ denotes parameters from training on the distilled dataset. Importantly, even when the model is trained on the distilled dataset $\hat{\mathbf{D}}$, we still evaluate its performance on the original $\mathbf{D}_{test}$ for a fair measurement. When creating the dataset $\hat{\mathbf{D}}$, the pairs $(\hat{x}, \hat{y})$ can be subsampled from the original set $\mathbf{D}$, as described in the coreset selection methods below (Sec. 3.2). We propose a much more effective strategy in Sec. 3.3 to learn *synthetic* image-text pairs $(\hat{x}, \hat{y})$, which can be more information-rich.

**Connection with Image-only Dataset Distillation.** Traditionally, dataset distillation is tailored for classification tasks with discrete labels, each of which possesses a distinctive set of distilled data that enables efficient learning while preserving important information. We take this concept a step further to the multimodal scenario, where we distill information from both vision and language data. This involves creating synthetic data that capture critical relationships within and between these two modalities. As opposed to merely classifying discrete labels, we are examining a more complex, interconnected dataset where the relation between modalities is crucial. Our method considers the image-text correlation and how they influence each other. It is worth noting that distillation would be impossible with single modality optimization (see Sec. 4.3).

## 3.2 Baselines: Coreset Selection

Since, to the best of our knowledge, there is no pre-existing work in the domain of vision-language dataset distillation, we begin by formulating a set of baselines to construct the smaller dataset $\hat{\mathbf{D}}$. These baselines are based on coreset selection methods, where a subset of the training pairs $(x_i, y_i)$ is chosen, up to a given budget of $M$ pairs, as to maximize the "informativeness" of the selected subset. We consider three such methods, adapted from prior work.

**Herding** (Welling, 2009) Herding selects data points based on the distance between the coreset center and the original dataset center in the feature space. It greedily adds one sample each time into the coreset to minimize the distance between two centers. We use pre-trained encoders to extract features from the image-text pairs, concatenate the features, and calculate the dataset center in the feature space by averaging all feature vectors. We start with an empty coreset and for each iteration, add the image-text pair that is closest to the current center of the coreset in Euclidean distance. We recalculate the coreset center after adding each data point.

**K-center** (Farahani & Hekmatfar, 2009; Sener & Savarese, 2017) Different from computing a single center in Herding, K-center selects the training examples that are maximally separated. Concretely, we concatenate the features of the image and text pairs and start by randomly selecting a single data point. Then, for each iteration, until K points are selected, we add a new image-text pair that is *furthest* in Euclidean distance from the nearest example. The drawback of this method is its high computational cost, especially with large datasets, as it involves heavy distance calculations between data points in each iteration.

**Forgetting** (Toneva et al., 2018) The core idea is to identify reliable training data that the original model consistently learns well. During each training epoch, we check how accurately the models predict every image-text pair for a specific task (i.e., image-text retrieval). A forgetting event is registered for an image-text pair when the model correctly predicts the data in one epoch but fails in the next. Throughout training, we continually track these forgetting events for each pair, to identify the ones with the *fewest* forgetting events.

### 3.3 Bi-trajectory Guided Vision-Language Co-Distillation

The coreset selection methods described above, while effective to some extent, demonstrate certain limitations as they only rely on selecting a subset of the training dataset $\mathbf{D}$. This restriction leads to less effective results compared to our method, as ours provides the flexibility to generate an optimized distilled dataset $\hat{\mathbf{D}}$, and the learning process efficiently helps extract the most essential information embedded in $\mathbf{D}$. Not only does this lead to decreased storage and computational requirements, but it also optimizes the performance of the model trained on this distilled dataset.

Here we describe our vision-language dataset distillation framework, building off of the idea of matching training trajectories (MTT) (Cazenavette et al., 2022) developed for distilling image classification datasets. The core idea of trajectory matching is that the dataset distillation can be achieved by implicitly matching the by-product, which is the parameter trajectory of the distilled dataset and the original full dataset, given direct information extraction is not feasible. We can compute a loss function on the cumulative discrepancy between the expert parameter trajectory $\theta^*$ obtained from the model trained on the full dataset $\mathbf{D}$ and the parameters $\hat{\theta}$ obtained from the model on the distilled dataset $\hat{\mathbf{D}}$, and use that loss to guide the creation of a better $\hat{\mathbf{D}}$, one that can match the parameters $\theta^*$ more closely. The approach consists of two stages:

1. Obtaining the expert training trajectories $\{\tau^*\}$, with each trajectory $\tau^* = \{\theta_t^*\}_{t=0}^T$, by training multiple models for $T$ epochs on the full dataset $\mathbf{D}$. For our multimodal setting, the models are trained using **bidirectional contrastive loss**, described below.

2. Training a set of student models on the current distilled dataset $\hat{\mathbf{D}}$ using the same bidirectional contrastive loss, and then updating $\hat{\mathbf{D}}$ based on the **bi-trajectory matching loss** of the student models' parameter trajectories and the expert trajectories $\tau^*$.

**Bidirectional Contrastive Loss.** We train both expert and student VLMs using the bidirectional contrastive loss, following the formulation of (Radford et al., 2021) as it is effective for learning shared image-text representation. Concretely, given a batch of $n$ image-text pairs $\{(x, y)\}$, either from the real dataset $\mathbf{D}$ or from the synthetic distilled dataset $\hat{\mathbf{D}}$, we jointly learn the encoders $f(x; \theta_{img})$ and $g(y; \theta_{txt})$ such that the cosine similarity of all $n$ correct image-text pairs is high and that of the $(n^2 - n)$ incorrect pairs is low. We define cosine similarity between image $x$ and text $y$ as: $\alpha_{xy} = \frac{\langle f(x;\theta_{img}), g(y;\theta_{txt}) \rangle}{\|f(x;\theta_{img})\|\|g(y;\theta_{txt})\|}$. We then compute bidirectional contrastive losses composed of an image-to-text matching loss and a text-to-image matching loss, following the form of the InfoNCE loss (Oord et al., 2018):

---

**Algorithm 1** Bi-Trajectory Co-Distillation

**Input:** $(\tau^*_{img}, \tau^*_{txt})$: expert trajectories over $T$ epochs, trained on $\mathbf{D}$. $M$: distilled dataset size. $T^+$: max start epoch. $R$: # updates between starting and target expert parameters. $\hat{R}$: # updates to student network per distillation step. $\alpha_0$: initial learning rate.

1: Initialize distilled data $\hat{\mathbf{D}} \sim \mathbf{D}$
2: Initialize trainable learning rate $\alpha := \alpha_0$ for training student models on $\hat{\mathbf{D}}$
3: **for each** distillation step **do**
4:     ▷ Randomly sample an expert trajectory $(\tau^*_{img}, \tau^*_{txt})$, corresponding to $\{(\theta^*_{img,t}, \theta^*_{txt,t})\}_{t=0}^T$
5:     ▷ Choose random start epoch, $s \leq T^+$
6:     ▷ Initialize the student network with expert params $\hat{\theta}_{img,s} := \theta^*_{img,s}$ and $\hat{\theta}_{txt,s} := \theta^*_{txt,s}$
7:     **for** $\hat{R}$ iterations **do**
8:         ▷ Sample a mini-batch of distilled dataset $\hat{\mathbf{D}}$
9:         ▷ Use the contrastive loss of Eqn. 3 to update the parameters $\hat{\theta}_{img}$ and $\hat{\theta}_{txt}$
10:     **end for**
11:     ▷ Compute loss $\ell_{trajectory}$ between $\theta^*_s$, $\theta^*_{s+R}$ and $\hat{\theta}_{s+\hat{R}}$ using Eqn. 4
12:     ▷ Update distilled dataset $\hat{\mathbf{D}}$ and learning rate $\alpha$ with respect to $\ell_{trajectory}$
13: **end for**
14: **Output:** Distilled data $\hat{\mathbf{D}}$ and learning rate $\alpha$.

---

$$\ell_{contrastive} = -\frac{1}{2n} \sum_{(x,y) \text{ in batch}} \left( \log \frac{\exp \alpha_{xy}}{\sum_{y' \neq y} \exp \alpha_{xy'}} + \log \frac{\exp \alpha_{xy}}{\sum_{x' \neq x} \exp \alpha_{x'y}} \right). \tag{3}$$

To imitate the effect of training data on parameter trajectories, we use the same objective function to guide the update of parameters $\theta_{img}, \theta_{txt}$ during both expert training (stage 1) and distillation (stage 2). Notably, while hard negative mining is typically used in conjunction with contrastive loss, here we rely fully on the dataset distillation process itself without additional intervention. This process inherently considers hard negatives; it distills samples that are hard negative samples for others, which are eventually effective samples for learning. Dataset distillation can potentially by-pass the traditional hard negative mining complexities through the learning process.

**Bi-Trajectory Matching Loss.** Following the formulation of MTT (Cazenavette et al., 2022) , we randomly sample $M$ image-text pairs from $\mathbf{D}$ to initialize the distilled dataset $\hat{\mathbf{D}}$ (more details can be found in the Sec 4.1). We sample an expert trajectory $\tau^* = \{\theta^*_t\}_{t=0}^T$ and a random starting epoch $s$ to initialize $\hat{\theta}_s = \theta^*_s$. We train the student model on the distilled dataset for $\hat{R}$ steps to obtain $\hat{\theta}_{s+\hat{R}}$. We then update the distilled dataset based on bi-trajectory matching loss $\ell_{trajectory}$ computed on the accumulated difference between student trajectory and expert trajectory:

$$\ell_{trajectory} = \frac{\|\hat{\theta}_{img,s+\hat{R}} - \theta^*_{img,s+R}\|_2^2}{\|\theta^*_{img,s} - \theta^*_{img,s+R}\|_2^2} + \frac{\|\hat{\theta}_{txt,s+\hat{R}} - \theta^*_{txt,s+R}\|_2^2}{\|\theta^*_{txt,s} - \theta^*_{txt,s+R}\|_2^2}. \tag{4}$$

We update the distilled dataset by back-propagating through multiple ($\hat{R}$) gradient descent updates to $\hat{\mathbf{D}}$, specifically in the image pixel space and text embedding space with respect to Eqn. 4. We initialize the continuous sentence embeddings using a pretrained BERT model and update the distilled text in the continuous embedding space. For the distilled image optimization, we directly update the pixel values of the distilled images. The full details are in Algorithm 1.

**Low-Rank Adaptation Matching.** Further, for complex image encoders like Vision Transformers (ViTs) (Dosovitskiy et al., 2020), the bi-trajectory matching does not work well due to the high dimensionality of the embeddings and large number of parameters saved in the trajectories compared to models like NFNet. To mitigate this issue, we propose Low-Rank Adaptation (LoRA) matching by matching the trajectories of only a small subset of the model's parameters through low-rank matrices. LoRA is effective for finetuning pretrained models (Hu et al., 2021), and it introduces trainable low-rank matrices to the weight matrices of specific layers of the pretrained models. LoRA matching optimizes the trajectories of low-rank adapters instead of the full parameters.

Given the weight matrix $W$ of certain layer in the ViT model, we introduce two low-rank matrices $A \in \mathbb{R}^{d \times r}$ and $B \in \mathbb{R}^{r \times d}$ to each layer's weight matrix $W$, where $d$ is the dimension of $W$ and $r \ll d$ represents the rank. The adaptation is performed by modifying $W$ to $W' = W + AB^T$, where $W'$ denotes the adapted weight matrix. We only train and save the weights from A and B in the expert trajectories and match the student trajectories with Eqn. 4. For example, the ViT model we used is `vit_base_patch16_224`, which has 86 million parameters, but with LoRA and $r = 4$, the parameters are reduced to 18 million, cutting 78.71% of the parameters. This allows for efficient adaptation of the model with minimal additional parameters. With LoRA matching, we can focus on a smaller set of parameters and efficiently optimize the $\ell_{trajectory}$ during distillation while maintaining the capacity to save critical information.

## 4 Experiments

In this section, we first describe the cross-modal retrieval test-bed in Sec. 4.1. We use it to evaluate our vision-language dataset co-distillation performance. We then compare our method to baseline coreset selection approaches and provide the key quantitative, qualitative results, and cross-architecture generalization results in Sec. 4.2. We further conduct a set of ablation studies in Sec. 4.3.

### 4.1 Vision-Language Distillation Setup

**Datasets and Tasks.** We evaluate our method on standard vision-language datasets: Flickr30K (Plummer et al., 2015) and COCO (Lin et al., 2014), which are widely used for image-text retrieval tasks. We use them for expert training (stage 1) and distillation (stage 2). We adopt the Karpathy split (Karpathy & Fei-Fei, 2015) for Flickr30K (29k/1k/1k) and COCO (113/5k/5k) for train/validation/test respectively. Each image is paired with five captions. We retrieve the closest matches using cosine distance from one modality based on a query from the other. We use R@K (for K$\in \{1, 5, 10\}$) to compute the fraction of times the correct result appears among the top K items. To move from distilling image-only datasets to vision-language datasets, we validate in appendix Sec. B if our method has potential in the classic image classification setting.

**Network Architectures.** We primarily use pretrained and trainable NormalizerFree ResNet (NFNet) (Brock et al., 2021b) as the image backbone following Flamingo (Alayrac et al., 2022) as well as Vision Transformer (ViT), for the text backbone we use pretrained and frozen BERT (Devlin et al., 2018). Ablation studies on different backbones are in Appendix Sec. E.2. While both the encoders are pretrained, they are only pretrained on unimodal data with *no* exposure to the other modality. Each encoder is followed by a trainable linear projection layer with random initialization. Using a trainable BERT adds additional complexity which is orthogonal to vision-language dataset distillation and is out of the scope of this work. Pretrained models serve as a common foundation and good starting point and see Appendix Sec. E.3 for details.

**Implementation.** For expert training, we train on a single RTX 3090 GPU for 10 epochs, where a single epoch takes 40 minutes of wall-clock time. Sampling from a set of trajectories encourages the distilled dataset to include diverse information and avoid overfitting to a particular step, thus we save 20 image-text bi-trajectories. For distillation, it takes 6 - 15 GPU hours depending on the settings (e.g. number of distilled pairs) with a 8-GPU A6000 node. We initialize a trainable learning rate $\alpha$ at 0.1 for the student model. We followed the data augmentation techniques in (Li et al., 2022), including resizing, cropping, flipping, and RandomAugment. We use SGD with momentum=0.5, the learning rate for updating $\alpha$, distilled image pixels, and distilled text embeddings are 1e-02, 1000, and 1000, respectively.

**Initialization.** Following prior studies (Nguyen et al., 2020; Zhou et al., 2022), we initialize the distilled set with randomly selected real samples. We randomly select $n \in \{100, 200, 500, 1000\}$ image-text pairs from the original dataset, with images at $224 \times 224$ resolution, and 768-dimensional sentence embeddings obtained via pretrained BERT. Our findings in Appendix Sec. E.1 show that initializing images from Gaussian distribution results in significantly lower performance. The complexity of images makes learning from random initialization challenging. In contrast, there is little difference in performance between using real and randomly initialized text embeddings. Surprisingly, despite the initial lack of semantic meaning between 'noise' texts and real images, we found notable semantic similarity between distilled text and real images, suggesting potential applications of our method in Visual Question Answering.

Table 1: **Baseline comparisons on Flickr30K (top) and COCO (bottom) with NFNet and BERT.** We compare our distillation method to four coreset selection methods: random selection of training examples (**R**), Herding (**H**) (Welling, 2009), K-center (**K**) (Farahani & Hekmatfar, 2009) and Forgetting (**F**) (Toneva et al., 2018). We consider different selected sizes (100, 200, 500, and 1000) and report the image-to-text (TR) and text-to-image (IR) R@1 retrieval performance on Flickr30K and COCO datasets. We report our distillation results along with standard deviation, they are calculated from the performance of five differently initialized models after training on the same distilled dataset. Full details with R@5/10 are in Appendix Tab. 6. In the very small budget regime, retrieval accuracy is much lower compared to the full data performance. As the budget grows, performance reaches full performance as shown in Tab. 8 in the appendix.

| | | TR | | | | | IR | | | | |
| | | Coreset Selection | | | | | Coreset Selection | | | | |
| Dataset | #pairs | R | H | K | F | Dist (ours) | R | H | K | F | Dist (ours) |
|---|---|---|---|---|---|---|---|---|---|---|---|
| Flickr30K | 100 | 1.3 | 1.1 | 0.6 | 1.2 | **9.9** $\pm$ **0.3** | 1.0 | 0.7 | 0.7 | 0.7 | **4.7** $\pm$ **0.2** |
| | 200 | 2.1 | 2.3 | 2.2 | 1.5 | **10.2** $\pm$ **0.8** | 1.1 | 1.5 | 1.5 | 1.2 | **4.6** $\pm$ **0.9** |
| | 500 | 5.2 | 5.1 | 4.9 | 3.6 | **13.3** $\pm$ **0.6** | 2.4 | 3.0 | 3.5 | 1.8 | **6.6** $\pm$ **0.3** |
| | 1000 | 5.2 | 5 | 5.6 | 3.1 | **13.3** $\pm$ **1.0** | 3.8 | 4.1 | 4.4 | 3.2 | **7.9** $\pm$ **0.8** |
| COCO | 100 | 0.8 | 0.8 | 1.4 | 0.7 | **2.5** $\pm$ **0.3** | 0.3 | 0.5 | 0.4 | 0.3 | **1.3** $\pm$ **0.1** |
| | 200 | 1.0 | 1.0 | 1.2 | 1.1 | **3.3** $\pm$ **0.2** | 0.6 | 0.9 | 0.7 | 0.6 | **1.7** $\pm$ **0.1** |
| | 500 | 1.9 | 1.9 | 2.5 | 2.1 | **5.0** $\pm$ **0.4** | 1.1 | 1.7 | 1.1 | 0.8 | **2.5** $\pm$ **0.5** |
| | 1000 | 1.9 | 2.4 | 2.4 | 1.9 | **6.8** $\pm$ **0.4** | 1.5 | 1.3 | 1.5 | 0.7 | **3.3** $\pm$ **0.1** |

## 4.2 Key Results

**Quantitative Results.** As shown in Tab. 1 and Tab. 6 in Appendix Sec. A, we observe that although there is relatively little variation in performance across each of the coreset selection baselines which we compare to, dataset distillation outperforms the best alternative by anywhere between 138% (improving R@1 from 5.6 of K-center (Farahani & Hekmatfar, 2009) to 13.3 of our model) to 661% (improving R@1 from 1.3 of random selection to 9.9 of our model). The relative improvement increases when fewer pairs are used for training.

Moreover, as shown in Tab. 6, we note that with 1000 pairs, almost 30 times fewer examples than in the original dataset, our data distillation approach reaches 43.7 R@10 for TR, relative to a practical upper bound of 75.2, and 34.4 for IR R@10, relative to an upper bound of 69.7. We also observe that the performance among the baseline coreset selection methods varies only slightly, with no single method consistently outperforming the others across all pair sizes and retrieval metrics, often matching or underperforming random selection. This suggests coreset selection limitations in multimodal settings. In comparison, our bi-trajectory co-distillation method is optimized for vision-language alignment settings and thus performs significantly better. Our results show the effectiveness of distilled data, achieving unparalleled efficiency with significantly fewer examples.

We compare the performance of the ViT model (`vit_base_patch16_224`) with and without the LoRA trajectory matching using BERT as the language encoder on the Flickr30K dataset in Tab. 2. Interestingly, vanilla ViT struggles in distillation, potentially due to attention mechanisms. For 100 pairs, the TR score jumps to 10.4 and the IR score to 5.4. With 1000 pairs, the improvement is even more noticeable: the TR score increases to 15.8 and the IR to 8.1. Those results show that the LoRA trajectory matching is much more effective for distilling critical information. We report the practical upper/lower performance in Tab. 3.

**Qualitative Results.** Here we provide distilled image-text pairs visualizations out of 100 distilled pairs from Flickr30K after 2000 distillation steps in Fig. 3. We visualize the distilled text embeddings via their nearest neighbor sentences (cosine similarity) in the training set embedding space for more intuitive understanding. Additional visualizations are in Appendix Sec. G. The distilled images, compared to the original ones, add high-frequency components that help improve the generalization performance (Wang et al., 2020a). While the distilled texts maintain semantic components associated with the distilled images and capture the key attributes e.g. `"couple"`, `"kiss"`, `"man"`, `"surf"`, `"huge wave"`, they also deviate from

Table 2: **Performance Comparisons of ViT with and without LoRA on Flickr30K (top) and COCO (bottom).** We report the distilled performance on Flickr30K and COCO using ViT with and without LoRA, we use BERT as the language encoder.

| Dataset | #Pairs | Without LoRA | | | | | | With LoRA | | | | | |
|---|---|---|---|---|---|---|---|---|---|---|---|---|---|
| | | TR | | | IR | | | TR | | | IR | | |
| | | R@1 | R@5 | R@10 | R@1 | R@5 | R@10 | R@1 | R@5 | R@10 | R@1 | R@5 | R@10 |
| Flickr30K | 100 | 1.5 | 2.5 | 4.5 | 0.6 | 1.2 | 2.3 | 10.4 | 23.6 | 38.7 | 5.4 | 18.8 | 27.4 |
| | 200 | 1.8 | 3.9 | 6.4 | 0.8 | 1.5 | 2.7 | 11.2 | 24.5 | 41.5 | 6.4 | 19.4 | 29.4 |
| | 500 | 2.1 | 4.3 | 7.2 | 1.5 | 2.1 | 3.6 | 13.4 | 27.8 | 43.4 | 7.6 | 21.1 | 32.7 |
| | 1000 | 3.3 | 5.8 | 7.9 | 1.5 | 2.3 | 3.9 | 15.8 | 29.7 | 45.9 | 8.1 | 23.4 | 35.8 |
| COCO | 100 | 0.5 | 0.9 | 2.1 | 0.3 | 0.7 | 1.4 | 5.1 | 17.4 | 27.1 | 2.3 | 8.1 | 14.5 |
| | 200 | 0.8 | 1.5 | 3.5 | 0.3 | 0.8 | 1.8 | 6.8 | 19.3 | 28.5 | 2.9 | 9.5 | 18.4 |
| | 500 | 1.2 | 2.3 | 4.1 | 0.5 | 1.1 | 2.3 | 7.4 | 21.4 | 29.4 | 3.8 | 11.2 | 19.6 |
| | 1000 | 1.5 | 2.7 | 4.5 | 0.7 | 1.5 | 2.9 | 9.9 | 22.5 | 32.8 | 4.7 | 12.7 | 20.2 |

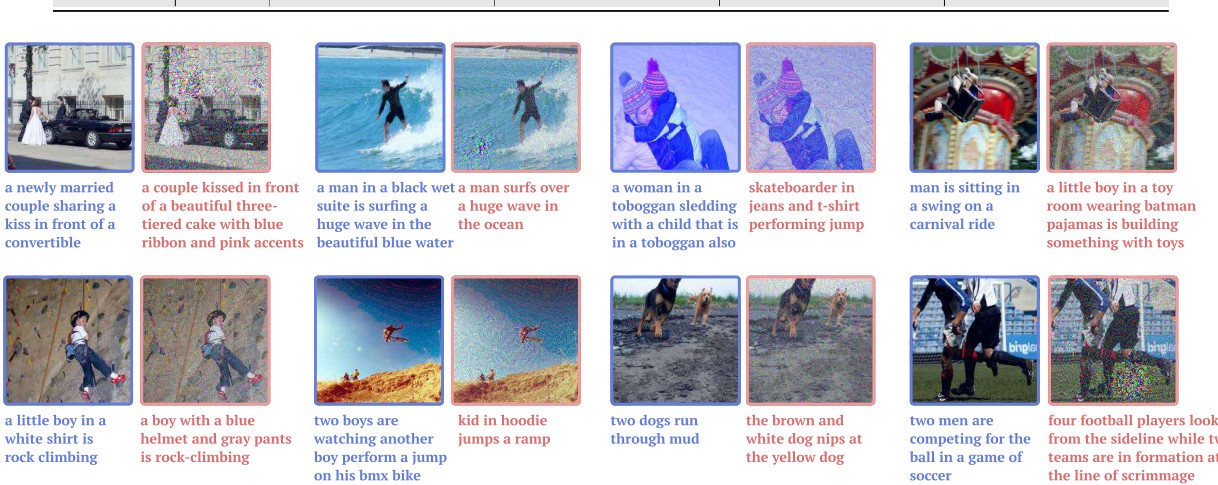

Figure 3: **Before and After Distillation.** (*Left*) The image-text pairs before the distillation. (*Right*) The image-text pairs after 2000 distillation steps. Note that the texts visualized here are the nearest sentence decodings in the training set of the distilled text embeddings.

Table 3: **Practical Limits Comparison.** A side-by-side comparison of the image-to-text (TR) and text-to-image (IR) retrieval results obtained from random ranking, full dataset training with NFNet and BERT, and full dataset training with ViT (with LoRA) and BERT on Flickr30K and COCO.

| Dataset | **Lower** Bound: Random Ranking | | | | | | **Upper** Bound: NFNet + BERT | | | | | | **Upper** Bound: ViT (LoRA) + BERT | | | | | |
|---|---|---|---|---|---|---|---|---|---|---|---|---|---|---|---|---|---|---|
| | TR | | | IR | | | TR | | | IR | | | TR | | | IR | | |
| | R@1 | R@5 | R@10 | R@1 | R@5 | R@10 | R@1 | R@5 | R@10 | R@1 | R@5 | R@10 | R@1 | R@5 | R@10 | R@1 | R@5 | R@10 |
| Flickr30K | 0.1 | 0.6 | 1.1 | 0.1 | 0.5 | 1.0 | 33.9 | 65.1 | 75.2 | 27.3 | 57.1 | 69.7 | 42.7 | 72.9 | 83.5 | 31.8 | 62.8 | 74.5 |
| COCO | 0.02 | 0.1 | 0.2 | 0.02 | 0.1 | 0.2 | 19.6 | 45.6 | 59.5 | 16.9 | 41.9 | 55.9 | 22.6 | 50.8 | 64.8 | 19.1 | 44.7 | 58.7 |

original sentence embeddings, as they are not in the original five captions paired with the images. The improved performance indicates that both high-frequency components and semantic ones are perceived by models and these significantly help in aligning vision-language modalities.

**Cross-Architecture Generalization.** Following previous works (Cazenavette et al., 2022; Cui et al., 2022; Zhao & Bilen, 2023), we evaluate the cross-architecture generalization ability of our distilled data in training unseen architectures. The experiments are conducted on Flickr30K with 100 distilled pairs. Distilling with NFNet model, we report the cross-architecture generalization performance on NF-ResNet50 (Brock et al.,

Table 4: **Cross-architecture Generalization.** Comparison of image-text retrieval performance when models are trained directly vs. transferred to the specified architecture. Evaluated on Flickr30K with 100 pairs.

| Distill | Evaluate | TR | | | IR | | |
|---|---|---|---|---|---|---|---|
| | | R@1 | R@5 | R@10 | R@1 | R@5 | R@10 |
| NFNet | NFNet | 9.9 | 28.3 | 39.1 | 4.7 | 15.7 | 24.6 |
| | NF-ResNet50 | 5.2 | 14.7 | 21.2 | 4.5 | 13.8 | 21.2 |
| | NF-RegNet | 3.6 | 9.7 | 15.5 | 2.5 | 8.6 | 14.0 |
| | ViT | 3.1 | 8.6 | 13.2 | 2.3 | 7.4 | 13.3 |

| Distill | Evaluate | TR | | | IR | | |
|---|---|---|---|---|---|---|---|
| | | R@1 | R@5 | R@10 | R@1 | R@5 | R@10 |
| ViT | ViT | 10.4 | 23.6 | 38.7 | 5.4 | 18.8 | 27.4 |
| | NF-ResNet50 | 2.8 | 8.3 | 12.2 | 2.0 | 6.7 | 11.5 |
| | NF-RegNet | 3.7 | 8.4 | 14.1 | 1.9 | 5.9 | 9.2 |
| | NFNet | 4.4 | 12.6 | 20.3 | 2.6 | 7.3 | 13.9 |

Table 5: **Ablation with single modality distillation**. We reported the image-text retrieval performance of text-only distillation (**T**), image-only distillation (**I**), and co-distillation (**Ours**) on Flickr30K. The results demonstrated the effectiveness of jointly distilling both image and text.

| | TR | | | | | | | | | IR | | | | | | | | |
|---|---|---|---|---|---|---|---|---|---|---|---|---|---|---|---|---|---|---|
| | R@1 | | | R@5 | | | R@10 | | | R@1 | | | R@5 | | | R@10 | | |
| # pairs | T | I | Ours | T | I | Ours | T | I | Ours | T | I | Ours | T | I | Ours | T | I | Ours |
| 100 | 1.3 | 3.5 | **9.9** | 3.5 | 11.5 | **28.3** | 5.9 | 17.4 | **39.1** | 0.5 | 1.6 | **4.7** | 2.1 | 5.6 | **15.7** | 3.4 | 9.7 | **24.6** |
| 200 | 1.4 | 4.5 | **10.2** | 4.8 | 12.8 | **28.7** | 8.2 | 21.7 | **41.9** | 0.7 | 2.0 | **4.6** | 2.7 | 8.1 | **16.0** | 4.7 | 13.0 | **25.5** |
| 500 | 6.6 | 6.5 | **13.3** | 19.5 | 19.4 | **32.8** | 30.4 | 28.9 | **46.8** | 3.8 | 3.8 | **6.6** | 13.5 | 12.4 | **20.2** | 20.8 | 19.9 | **30.0** |
| 1000 | 7.7 | 5.0 | **13.3** | 20.7 | 17.4 | **34.8** | 31.2 | 24.9 | **45.7** | 4.0 | 3.9 | **9.1** | 13.3 | 13.1 | **24.1** | 20.1 | 20.1 | **33.8** |

2021a), NF-RegNet (Xu et al., 2022), and ViT (Dosovitskiy et al., 2020) (LoRA). As shown in Tab. 4, our method transfers well across different models.

## 4.3 Ablation Studies

We conduct a set of ablation studies to understand unimodal distillation vs. co-distillation, distilled dataset initialization (Sec. E.1), different encoder backbones (Sec. E.2), pretraining (Sec. E.3), synthetic steps (Sec. E.4), and their influence on distillation.

We compare co-distillation with unimodal distillation, where we keep one of the modalities fixed during distillation. Tab. 5 shows the retrieval performance of text-only distillation, image-only distillation, and co-distillation. Across all tasks and metrics, the co-distillation approach clearly outperforms the others. We observed that the performance of text-only distillation is worse than that of image-only distillation. This may not be surprising: text descriptions typically contain only a salient but small portion of visual information. However, descriptions in the evaluated datasets typically contain no information that cannot be inferred from the images. By distilling images to text-relevant aspects, it can highlight essential image features. Thus, if we interpret each original image as having substantially more information than its original sentence, we would expect image-only distillation to perform better in a smaller-scale regime (removing spurious information) and text-only distillation to perform better in a larger-scale regime (adding useful details).

In contrast, co-distillation allows the synthetic dataset to further optimize for compact representation and efficient storage, removing redundant information between examples in the smaller-scale contexts and adding information not present in the selected original images in larger-scale contexts. Our co-distillation method, combining text and image modalities during training, consistently outperforms single-modality distillation across different numbers of training pairs and metrics. While the improvement from co-distillation is consistent, it is particularly substantial with fewer pairs: in the 100 and 200 pairs rows, co-distillation outperforms its unimodal alternatives by over $2\times$. In fact, co-distillation with 100 pairs consistently outperforms unimodal distillation with 1000 pairs. These results demonstrate the effectiveness of jointly distilling across modalities and highlight the complementary nature of multimodal data.

## 5    Conclusion

In this work, we propose the first vision-language dataset distillation method. By co-distilling both vision and language modalities, we can progressively optimize and distill the most critical information from a vision-language dataset. Our experiments show that co-distilling different modalities via bi-trajectory matching and using LoRA matching for complex model finetuning hold promise. We hope that the insights we gathered can serve as a roadmap for future studies exploring more complex settings. Furthermore, we believe our work lays the groundwork for future research aimed at understanding the minimum information required for a vision-language model to achieve comparable performance quickly, thereby building a better understanding of the compositionality of compact visual-linguistic knowledge.

**Limitations.** We make note of two limitations of our approach. Firstly, dataset distillation is not exempt from the "No Free Lunch" theorem (Wolpert & Macready, 1997). As discussed in (Sachdeva & McAuley, 2023), we also observed that the effectiveness of the distilled data is highly influenced by learning algorithms and models used during distillation, which could potentially lead to poor transferability. Furthermore, many dataset distillation methods are computationally intensive, i.e. the bi-level optimization in meta-learning distillation approaches, which is another major challenge. In contrast, our trajectory matching approach is significantly less computationally demanding, yet we observed that the larger synthetic steps often result in improved performance, and exploring closed-form solutions, i.e. implicit gradient-based methods (Lorraine et al., 2020) could be promising future directions to pursue.

**Broader Impact Statement.** Our exploration focuses on scientific understanding and practical applications of vision-language dataset distillation. While our work does not directly imply negative impacts, it may indirectly propagate existing biases in the original datasets. Therefore, it is important to incorporate rigorous bias-mitigation measurements for dataset distillation. Discussion on these critical aspects should remain a priority as we further explore the potential of vision-language dataset distillation.

## Acknowledgements

This material is based upon work supported by the National Science Foundation under Grants No. 2107048 and No. 2112562. Any opinions, findings, conclusions, or recommendations expressed in this material are those of the author(s) and do not necessarily reflect the views of the National Science Foundation. We thank many people from Princeton Visual AI lab (Allison Chen, Jihoon Chung, Tyler Zhu, Ye Zhu, William Yang and Kaiqu Liang) and Princeton NLP group (Carlos E. Jimenez, John Yang), as well as Tiffany Ling, George Cazenavette and Ilia Sucholutsky for their helpful feedback.

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

**Appendix**

In this Appendix, we first we provide the full baseline comparison (with R@1/5/10) on Flickr30K and COCO (Sec. A). Then we show the challenges of vision-language distillation (Sec. B) by transitioning the trajectory-matching pipeline from image-only to image-text retrieval. We provide analysis on distilled images (Sec. D) and lossless distillation (Sec. C). We further extend the ablation study, analyzing components of our pipeline, i.e. distilled dataset initialization (Sec E.1), encoder backbones (Sec. E.2), pretraining (Sec. E.3) and synthetic steps (Sec. E.4). Lastly, we show additional visualizations of the distilled samples, as well as the ones under different backbones (Sec. G).

## A  Full Details for Distilled Performance

We provide full distillation results following Section 4.2, including image-to-text and text-to-image retrieval results R@5 and R@10 with NFNet in Tab. 6.

Table 6: **Detailed Baseline comparisons of NFNet on Flickr30K (top) and COCO (bottom).** Following Tab. 1, here we report the full details of the distilled performance on Flickr30K and COCO with NFNet and BERT.

| Dataset | #pairs | Metrics | TR | | | | | IR | | | | |
| | | | Coreset Selection | | | | Dist (ours) | Coreset Selection | | | | Dist (ours) |
| | | | R | H | K | F | | R | H | K | F | |
| Flickr30K | 100 | R@1 | 1.3 | 1.1 | 0.6 | 1.2 | **9.9 $\pm$ 0.3** | 1.0 | 0.7 | 0.7 | 0.7 | **4.7 $\pm$ 0.2** |
| | | R@5 | 5.9 | 4.7 | 5.0 | 4.2 | **28.3 $\pm$ 0.5** | 4.0 | 2.8 | 3.1 | 2.4 | **15.7 $\pm$ 0.5** |
| | | R@10 | 10.1 | 7.9 | 7.6 | 9.7 | **39.1 $\pm$ 0.7** | 6.5 | 5.3 | 6.1 | 5.6 | **24.6 $\pm$ 1.0** |
| | 200 | R@1 | 2.1 | 2.3 | 2.2 | 1.5 | **10.2 $\pm$ 0.8** | 1.1 | 1.5 | 1.5 | 1.2 | **4.6 $\pm$ 0.9** |
| | | R@5 | 8.7 | 8.4 | 8.2 | 8.4 | **28.7 $\pm$ 1.0** | 4.8 | 5.5 | 5.4 | 3.1 | **16.0 $\pm$ 1.6** |
| | | R@10 | 13.2 | 14.4 | 13.5 | 10.2 | **41.9 $\pm$ 1.9** | 9.2 | 9.3 | 9.9 | 8.4 | **25.5 $\pm$ 2.6** |
| | 500 | R@1 | 5.2 | 5.1 | 4.9 | 3.6 | **13.3 $\pm$ 0.6** | 2.4 | 3.0 | 3.5 | 1.8 | **6.6 $\pm$ 0.3** |
| | | R@5 | 18.3 | 16.4 | 16.4 | 12.3 | **32.8 $\pm$ 1.8** | 10.5 | 10 | 10.4 | 9.0 | **20.2 $\pm$ 1.2** |
| | | R@10 | 25.7 | 24.3 | 23.3 | 19.3 | **46.8 $\pm$ 0.8** | 17.4 | 17.0 | 17.3 | 15.9 | **30.0 $\pm$ 2.1** |
| | 1000 | R@1 | 5.2 | 5 | 5.6 | 3.1 | **13.3 $\pm$ 1.0** | 3.8 | 4.1 | 4.4 | 3.2 | **7.9 $\pm$ 0.8** |
| | | R@5 | 15.6 | 14.6 | 16.1 | 14.9 | **34.8 $\pm$ 1.9** | 11.8 | 12.1 | 12.8 | 9.5 | **24.1 $\pm$ 1.6** |
| | | R@10 | 21.4 | 20.4 | 20.8 | 18.9 | **45.9 $\pm$ 2.5** | 19.9 | 20.0 | 20.4 | 18.7 | **33.8 $\pm$ 2.0** |
| COCO | 100 | R@1 | 0.8 | 0.8 | 1.4 | 0.7 | **2.5 $\pm$ 0.3** | 0.3 | 0.5 | 0.4 | 0.3 | **1.3 $\pm$ 0.1** |
| | | R@5 | 3.0 | 2.1 | 3.7 | 2.6 | **10.0 $\pm$ 0.5** | 1.3 | 1.4 | 1.4 | 1.5 | **5.4 $\pm$ 0.3** |
| | | R@10 | 5.0 | 4.9 | 5.5 | 4.8 | **15.7 $\pm$ 0.4** | 2.7 | 3.5 | 2.5 | 2.5 | **9.5 $\pm$ 0.5** |
| | 200 | R@1 | 1.0 | 1.0 | 1.2 | 1.1 | **3.3 $\pm$ 0.2** | 0.6 | 0.9 | 0.7 | 0.6 | **1.7 $\pm$ 0.1** |
| | | R@5 | 4.0 | 3.6 | 3.8 | 3.5 | **11.9 $\pm$ 0.6** | 2.3 | 2.4 | 2.1 | 2.8 | **6.5 $\pm$ 0.4** |
| | | R@10 | 7.2 | 7.7 | 7.5 | 7.0 | **19.4 $\pm$ 1.2** | 4.4 | 4.1 | 5.8 | 4.9 | **12.3 $\pm$ 0.8** |
| | 500 | R@1 | 1.9 | 1.9 | 2.5 | 2.1 | **5.0 $\pm$ 0.4** | 1.1 | 1.7 | 1.1 | 0.8 | **2.5 $\pm$ 0.5** |
| | | R@5 | 7.5 | 7.8 | 8.7 | 8.2 | **17.2 $\pm$ 1.3** | 5.0 | 5.3 | 6.3 | 5.8 | **8.9 $\pm$ 0.7** |
| | | R@10 | 12.5 | 13.7 | 14.3 | 13.0 | **26.0 $\pm$ 1.9** | 8.7 | 9.9 | 10.5 | 8.2 | **15.8 $\pm$ 1.5** |
| | 1000 | R@1 | 1.9 | 2.4 | 2.4 | 1.9 | **6.8 $\pm$ 0.4** | 1.5 | 1.3 | 1.5 | 0.7 | **3.3 $\pm$ 0.1** |
| | | R@5 | 7.6 | 9.0 | 9.0 | 7.7 | **21.9 $\pm$ 1.2** | 5.6 | 5.7 | 7.1 | 4.6 | **11.9 $\pm$ 0.5** |
| | | R@10 | 12.7 | 14.0 | 14.1 | 13.0 | **31.0 $\pm$ 1.5** | 9.6 | 10.1 | 10.9 | 8.0 | **22.1 $\pm$ 0.9** |

# B  CIFAR10 Classification vs Retrieval Distillation

Prior work has shown remarkable distillation results on CIFAR10 (Krizhevsky et al., 2009) classification. To move from distilling image-only datasets to vision-language datasets, we first check if our method has potential in simple settings. Concretely, we convert CIFAR10 labels to captions that pair with their corresponding images. Under this formulation, the objective of classification is equivalent to that of image-to-text retrieval (TR): finding the best text given an image.

In Tab. 7, we compare CIFAR10 distillation performance for dataset size of 1, 10, 50 images per class (IPC), under three different settings: classification, single-caption retrieval, and multi-caption retrieval. For classification, we demonstrate results from MTT (Cazenavette et al., 2022), where they distill an image-only dataset using expert trajectories trained on image-label pairs. In single-caption TR, we distill image-caption pairs using expert trajectories trained when each image is paired with a single caption `"This is a {label}"`. In multi-caption TR, we distill image-caption pairs but the expert trajectories are trained when each image is paired with five captions that are generated with varies prompts from (Radford et al., 2021). For consistency, all image trajectories are obtained with the 3-layer ConvNet backbone as specified in (Cazenavette et al., 2022), and text trajectories are from linear projection layers over pretrained BERT (Devlin et al., 2018) embeddings. Although the performance of vision-language distillation trails behind that of image-only distillation, the gap closes at larger IPCs. However, this gap highlights the challenge of the continuous label space in vision-language datasets. Moreover, the performance gap between single and multi-caption retrieval demonstrates the challenge of capturing the variability within human language descriptions.

Table 7: **CIFAR10 Classification vs Retrieval.** We provided ipc=1/10/50 classification performance vs. image-to-text retrieval R@1, which both measure whether an image has been matched with the correct class.

| IPC | Classification | image-to-text retrieval | |
|---|---|---|---|
| | | Single Caption | Multi Caption |
| 1 | $46.3 \pm 0.8$ | $27.4 \pm 1.0$ | $22.3 \pm 1.0$ |
| 10 | $65.3 \pm 0.7$ | $35.9 \pm 0.7$ | $33.2 \pm 0.5$ |
| 50 | $71.6 \pm 0.2$ | $66.8 \pm 1.1$ | $62.0 \pm 0.8$ |
| Full | $84.8 \pm 0.1$ | $79.6 \pm 0.6$ | $80.3 \pm 0.4$ |

# C  Upper Bound Performance

We further increase the distilled size to be 10% of the original Flickr30K dataset size and we provide the comparisons for distillation performance with the upper bound results (Tab. 8). The distillation performance are closely approaching the upper bound results.

Table 8: **Matching Upper Bound Performance.** With only 10% of the original size of Flickr30K, models trained on this distilled data show remarkable performance, closely approaching the upper bound results. For certain metrics, such as NFNet + CLIP with Text Retrieval (TR) at R@1, they reach as high as 98% of the upper bound.

| Result Type | Vision Backbone | Language Backbone | Ratio | TR | | | IR | | |
|---|---|---|---|---|---|---|---|---|---|
| | | | | R@1 | R@5 | R@10 | R@1 | R@5 | R@10 |
| Distillation | NFNet | BERT | 10% | 32.1 | 60.0 | 73.2 | 24.1 | 53.9 | 66.5 |
| Upper Bound | NFNet | BERT | 100% | 33.9 | 65.1 | 75.2 | 27.3 | 57.2 | 69.7 |
| Distillation | NFNet | CLIP | 10% | 60.0 | 86.3 | 91.4 | 47.4 | 78.2 | 86.5 |
| Upper Bound | NFNet | CLIP | 100% | 61.2 | 87.5 | 92.8 | 49.8 | 79.8 | 88.3 |

# D    Analysis on Distilled Images

We have found that increasing the learning rate and distillation time lead to more noticeable changes in the images within the distilled dataset (distilled images: Fig. 4, original images: Fig. 5). However, it is important to note that a higher learning rate or longer distillation time does not necessarily translate to improved performance of the distilled dataset, even if the images appear to deviate more drastically from the human perception perspective. Changes in image pixels alone may not reliably predict distillation performance. It is rather a measurement of the distillation strength. More distorted images suggest uneven pixel updates, while even updates yield results similar to the visualization we provided before in Fig. 3.

In line with previous studies, we initially expected more obvious changes in images would lead to better performance, but our findings suggest a different behavior of vision-language distillation with trajectory matching framework, reflecting how models capture vision-language interaction. From a human perception perspective, the distilled images appear to be moving less compared to previous classification works, yet those small vectors are still meaningful and contain useful information, as opposed to artifacts like noisy patterns. Our algorithm achieves a clear and consistent improvement over random baselines indicated by the results. We hope this discussion can inspire more research on vision-language dataset distillation.

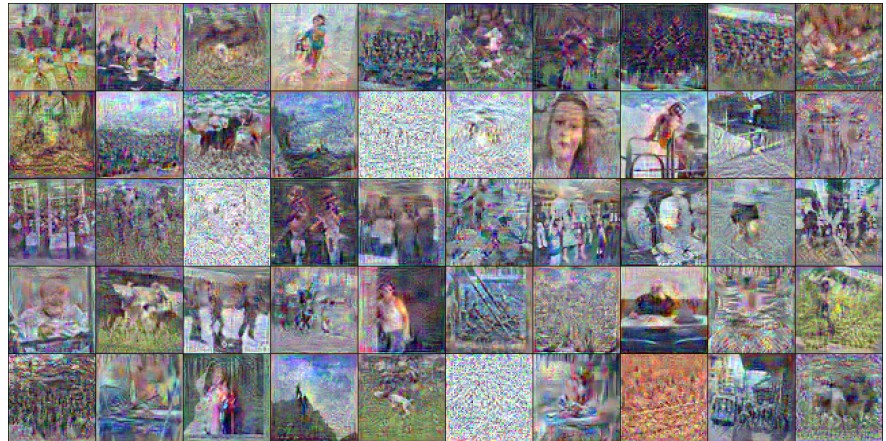

Figure 4: Distilled Images, iteration = 7000, lr image = 5000.

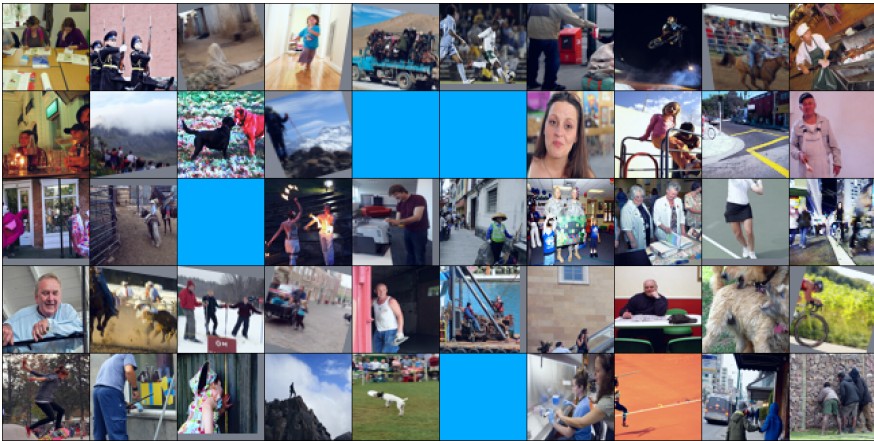

Figure 5: Original Images, iteration = 0.

# E   Additional Ablation Studies

In this section, we provide additional ablation studies. Unless specified, these distillation experiments are conducted on the Flickr30K dataset to distill 100 image-text pairs, and we use pretrained NFNet and BERT as backbones, with synthetic step set to 8 during distillation.

## E.1   Distilled Dataset Initialization

In the main paper, we provided experiments with real sample initialization. Here we experiment and evaluate initializing with Gaussian noise. Our findings in Tab. 9 show that initializing images from the Gaussian distribution results in significantly lower performance. It is worth noting that the complexity of images, which encodes a high degree and rich information of colors, shapes, textures and spatial relationships between objects, can make it difficult for models to learn effectively from randomly initialized images. On the other hand, using real text sampled from the training set vs. randomly initialized text embeddings does not bring a significant difference. We assume that the pretrained language models are good at generating or transforming 'noise' text embedding into meaningful sentences during the learning process, partly due to the inherent structure and predictability of language. We provide visualizations of real images and 'noise' texts combination below in Fig. 6 and Fig. 7 and Tab. E.1. To our surprise, even though the initialized 'noise' texts are not semantically meaningful to the initialized real images, we discovered a substantial degree of semantic similarity between the initialized real images and the learned distilled text. This suggests the probability of future application of our method in Visual Question Answering (VQA).

Table 9: **Image-Text Pair Initialization.** We compare the retrieval performance achieved with different combinations of image and text initialization strategies. The ✓denotes the use of real images or texts directly sampled from the training set, otherwise indicates the use of randomly initialized image or text, following Gaussian distribution. We can see that if we initialize the image from scratch, the performance will be pretty low. On the contrary, the performance did not drop too much if we start with 'noise' texts and real images, which indicates the importance of image signal for the small distilled set.

| | | Distillation | | | | | |
| | | TR | | | IR | | |
| Real Image | Real Text | R@1 | R@5 | R@10 | R@1 | R@5 | R@10 |
|---|---|---|---|---|---|---|---|
| ✓ | ✓ | **9.9** | **28.3** | 39.1 | **4.7** | **15.7** | **24.6** |
| ✓ | | 9 | 27.2 | **40.1** | 3.9 | 13.2 | 20.6 |
| | ✓ | 0.2 | 0.7 | 1.1 | 0.1 | 0.5 | 1 |
| | | 0.1 | 0.3 | 0.4 | 0.1 | 0.4 | 0.8 |

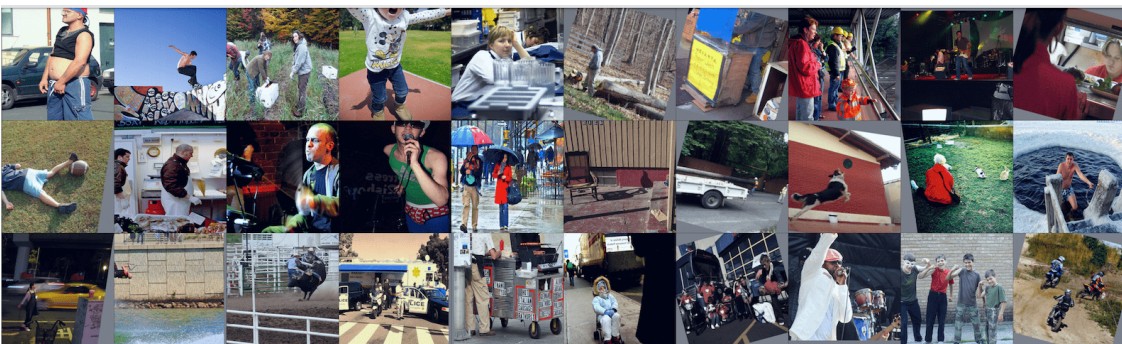

Figure 6: Initialized Images, iteration = 0.

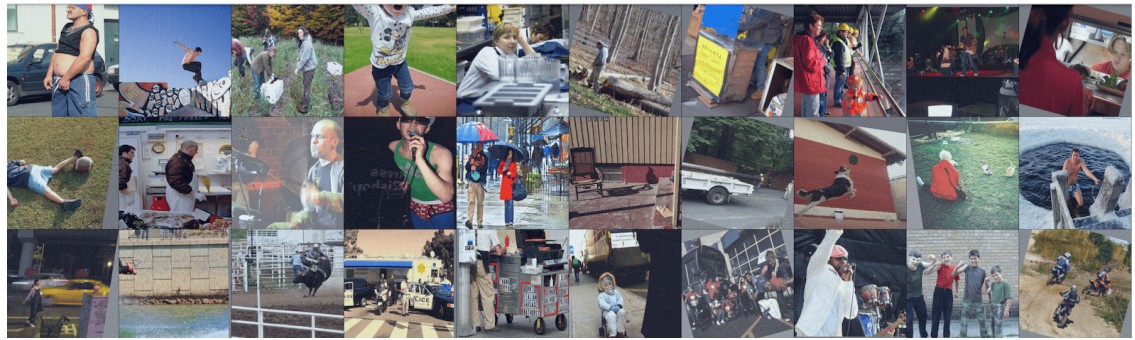

Figure 7: Distilled Images, iteration = 1000.

**'Noise' Texts, iteration = 0.**
30 randomly initialized text
from Gaussian distribution, we use nearest neighbor to find their closest sentence in the training set in Flickr30k for visualization purposes.

```
this man is fit and well toned running enthusiast
the music concert is just started at the giant stadium
a man in a beige shirt and tan slacks sits in a chair next to a hospital patient wearing a blue gown who is sitting
cross-legged on his hospital bed
man and woman employed by mongolian barbecue stand at counter
woman cupping water with hands over bathroom sink as child stands beside her
near snowflake sign, man sits while another stands wearing badge and headphones
very brave snow skier doing a flip off a cliff
a guy, dressed nicely, is painting a mural on a wall, with a ladder sitting beside him
dog chasing brown cow and black cow
seems to me looks like people in a work room or office working they all using laptop computers from apple it seems there dr
pepper soda and water bottle on
olympian performing on the rings
man walking behind distracted-looking woman carrying bags and camera
three men in caps sit at fireside near cabin, reading at night
the dog with the red collar is white, black, and brown
minor league pitcher
man in chair laughing and talking to others, while handling books
the man, with no shirt, reaches into a bucket to extract the substance inside small brown dog on leash
woman sitting at a park bench reading a book
violin soloists take the stage during the orchestra's opening show at the theater
black dog sitting while eating with neon yellow band around shoulders
several people are standing under a tarp two ladies are facing each other and one has a backpack on with her hands in her
jeans pockets while the other one
a boy wearing a flowered shirt raises his arm and jumps
a quarterback is looking to set up a pass from the end zone, while a teammate provides some blocking
a woman in a red coat takes a picture near marble columns at twilight
people with anti-immigration signs
the outside of a restaurant called el triuneo
cheerleaders build a pyramid near the goal-line
baby wears green frog big and makes grotesque face
a yellow, suspended roller coaster on a yellow track is midway through a loop
```

## E.2 Encoder Backbone Selection

In this section, we evaluate the impact of different language/vision backbones on the distillation performance.

### E.2.1 Language Backbones

Perhaps not surprisingly, CLIP (Radford et al., 2021) text encoder significantly outperforms BERT in all evaluation metrics, with a striking peak performance in TR R@10 at 92.8% for expert training. This exceptional performance can be mainly attributed to the fact that the pre-trained, off-the-shelf CLIP model is designed to learn a shared embedding space across multi-modalities. Although CLIP also shows a performance drop during distillation, it still retains a relatively high performance recovery ratio. In Sec. G we provide visualization of synthetic data distilled via NFNet and CLIP.

### E.2.2 Vision Backbones

The vision encoders carry the main gradient flows for the distillation process. We experimented on several vision backbones, and found that the architecture choice strongly influences the distillation quality. Similar

**Distilled Texts, iteration = 1000.**
Starting with randomly initialized text from Gaussian distribution, here is the synthetic text after distillation.

```
superhero man leaping in a plaza
a guy in a blue shirt listens to music as he skateboards along the edge of a ramp
tiger woods about to make a putt
little boy pulling a green wagon wearing a sweatshirt and boots
a young girl with blond-hair and glasses sitting at a table in a restaurant
three black young man are working in a semi-deserted area with a pile of construction material and jugs, one of them is
digging
woman buying cups of fruit from street vendor
six men in blue jumpsuits and a man in an orange jumpsuit walk by a shipyard
young girl balances on a reclined man's legs as part of a performance in front of an audience
a woman fillets a fish, as part of preparing a recipe that includes broccoli, celery, and eggs
a young man wearing a white shirt and red shorts kicking a ball
contortionist in strange checkered outfit wearing a white mask
one man plays an acoustic guitar, while another accompanies him on the accordion
male wearing brown shirt holding a microphone with an expression of singing
a young lady in a colorful dress, holds a white stuffed animal stands in the rain hold a plaid umbrella
a person with blue and polka-dot socks jumps on a bed with a red and white blanket
a damaged black color car on the street
skateboarder jumping in air and his skateboard is between his legs
a woman with a guitar sings in front of a building and grass
two woman are sitting on a beach together, facing the water
a busy street with building lined up and people walking down the street outside and nighttime
parasailer doing flip in midair
crowded arena with lots of people wearing yellow, carrying red flags
men in turbans laying down and examining cloth
a woman in a white apron prepares various meats on a large grill
a middle-aged man in a trench coat sleeps on a bus or train
a line of people, some standing and some sitting, are waiting on a platform for a train
three men playing drums, bass, and piano
a dirt-blonde girl in a white top with a key necklace holds a bag, standing in front of a sidewalk of street
cattle-drawn wagons traveling down a paved road and loaded with sticks
```

Table 10: **Ablation Analysis on Language Backbones.** We provide expert training and distillation performance evaluation for both pretrained BERT and CLIP models. CLIP text encoder demonstrates a strong capacity for high-recall retrieval.

| Language Model | Expert | | | | | | Distillation | | | | | |
|---|---|---|---|---|---|---|---|---|---|---|---|---|
| | TR | | | IR | | | TR | | | IR | | |
| | R@1 | R@5 | R@10 | R@1 | R@5 | R@10 | R@1 | R@5 | R@10 | R@1 | R@5 | R@10 |
| BERT | 33.9 | 65.1 | 75.2 | 27.3 | 57.2 | 69.7 | 9.9 | 28.3 | 39.1 | 4.7 | 15.7 | 24.6 |
| CLIP | **61.2** | **87.5** | **92.8** | **49.8** | **79.8** | **88.3** | **31.4** | **58.8** | **72.0** | **17.1** | **41.9** | **56.2** |

to dataset distillation by gradient matching (Zhao & Bilen, 2021b), batch normalization has an impact on the gradient/parameter matching framework. This is mainly because batch normalization incorporates a non-parametric component that can only be accumulated with batches and can not be trained.

Table 11: **Ablation Analysis on Vision Backbones.** We provide an extensive evaluation of several pretrained vision backbones including NFNet_l0 (Brock et al., 2021b), NF_ResNet50 (Brock et al., 2021a), NF_RegNet (Xu et al., 2022), and ResNet50 (He et al., 2016). This underscores the influence of architecture and highlights the potential negative impact of batch normalization for distillation.

| Vision Model | Expert | | | | | | Distillation | | | | | |
|---|---|---|---|---|---|---|---|---|---|---|---|---|
| | TR | | | IR | | | TR | | | IR | | |
| | R@1 | R@5 | R@10 | R@1 | R@5 | R@10 | R@1 | R@5 | R@10 | R@1 | R@5 | R@10 |
| ViT (LoRA) | **40.7** | **69.8** | **80.1** | **28.8** | **59.3** | **73.4** | **10.4** | 23.6 | 38.7 | **5.4** | **18.8** | **27.4** |
| NFNet-l0 | 33.9 | 65.1 | 75.2 | 27.3 | 57.2 | 69.7 | 9.9 | **28.3** | **39.1** | 4.7 | 15.7 | 24.6 |
| NF_ResNet50 | 28.9 | 56.6 | 71 | 22.8 | 50.1 | 63.4 | 6.5 | 18.2 | 28.1 | 3.5 | 11.6 | 18.7 |
| NF_RegNet | 26.9 | 57.2 | 70.2 | 21.1 | 50.1 | 62.9 | 7.8 | 21.9 | 33.3 | 3.3 | 12.7 | 20.5 |
| ResNet50 | 18 | 43.5 | 59.5 | 13.4 | 36.6 | 49.9 | 0.5 | 2.4 | 3.8 | 0.3 | 1.6 | 3.6 |

### E.3 Pretrained vs. Non-pretrained

Tab. 12 demonstrates the pretraining influence of the backbone encoders. Optimal performance is observed when both language and vision backbones are pretrained. This emphasizes the importance of pretraining before the expert training stage for large models and datasets.

Table 12: **Pretraining Impact.** Expert performance comparison for different pretraining configurations of vision and language backbones. The checkmark (✓) indicates the model was pretrained.

| Language Backbone | Vision Backbone | Expert | | | | | |
|---|---|---|---|---|---|---|---|
| | | TR | | | IR | | |
| | | R@1 | R@5 | R@10 | R@1 | R@5 | R@10 |
| ✓ | ✓ | **33.9** | **65.1** | **75.2** | **27.3** | **57.2** | **69.7** |
| ✓ | | 4.4 | 14.1 | 20.7 | 3.5 | 11.4 | 18.8 |
| | ✓ | 0.5 | 1.1 | 1.8 | 0.3 | 0.7 | 1.4 |
| | | 0.3 | 1 | 1.5 | 0.1 | 0.7 | 1.3 |

### E.4 Synthetic Steps

The synthetic step size plays an important role in optimizing the dataset distillation performance, as shown in Tab. 13. Using larger synthetic steps tends to achieve better distillation performance.

Table 13: **Synthetic Steps Impact**. Larger synthetic steps greatly improve performance. For 100 pairs with a synthetic step of 1, the performance is even below random selection. Setting the synthetic steps to a low value typically takes longer to optimize the distilled set and it is challenging with very small sets (e.g., # Pairs=100).

| #Pairs | #Syn Steps | Distillation | | | | | |
|---|---|---|---|---|---|---|---|
| | | TR | | | IR | | |
| | | R@1 | R@5 | R@10 | R@1 | R@5 | R@10 |
| 100 | 1 | 0.5 | 2.1 | 4.4 | 0.3 | 1.5 | 2.8 |
| | 2 | 7.1 | 23.4 | 32.9 | 3.0 | 10.2 | 16.4 |
| | 4 | 8.2 | 24.9 | 35.2 | 3.5 | 12.2 | 20.7 |
| | 8 | **9.9** | **28.3** | **39.1** | **4.7** | **15.7** | **24.6** |
| 200 | 1 | 3.2 | 9.3 | 14.1 | 1.6 | 5.2 | 8.8 |
| | 2 | 6.5 | 19.2 | 29.1 | 1.6 | 5.9 | 10.0 |
| | 4 | 8.2 | 24.5 | 34.4 | 2.2 | 7.4 | 11.8 |
| | 8 | **10.2** | **28.7** | **41.9** | **4.6** | **16.0** | **25.5** |
| 500 | 1 | 6.6 | 18.1 | 25.5 | 2.1 | 10.1 | 16.3 |
| | 2 | 8 | 21.7 | 31.3 | 3.8 | 14.9 | 23.2 |
| | 4 | 8.1 | 23.6 | 34.9 | 4.4 | 15.2 | 23.7 |
| | 8 | **13.3** | **32.8** | **46.8** | **6.6** | **20.2** | **30.0** |
| 1000 | 1 | 7.3 | 20.6 | 29.7 | 3.9 | 13.2 | 20.7 |
| | 2 | 8.8 | 26.8 | 36.6 | 5.7 | 17.4 | 26.4 |
| | 4 | 10.4 | 29.1 | 37.9 | 6.6 | 19.5 | 29.5 |
| | 8 | **13.3** | **34.8** | **45.7** | **9.1** | **24.1** | **33.8** |

## F  Beyond Trajectory Matching

In this section, we further provide experiment results of a distribution matching (Zhao & Bilen, 2023) baseline adapted to the vision-language setting. To use distribution matching for vision-language dataset distillation, concretely, we minimize the maximum mean discrepancy (mmd) between two distributions by sampling NFNet with different initialization and pretrained BERT. Similar to the distribution matching setting for image classification, we update the distilled data via mmd for vision and language modalities to match the original data distribution in a family of embedding spaces. We provide the comparison of Our method w/ DM (distribution matching) and Our method w/ TM (trajectory matching) on Flickr30K (R@1) in Tab. 14.

| # pairs | TR | | IR | |
|---|---|---|---|---|
| | **Ours w/ DM** | **Ours w/ TM** | **Ours w/ DM** | **Ours w/ TM** |
| 100 | $3.2 \pm 1.8$ | $\mathbf{9.9} \pm \mathbf{0.3}$ | $1.4 \pm 0.7$ | $\mathbf{4.7} \pm \mathbf{0.2}$ |
| 200 | $3.3 \pm 1.3$ | $\mathbf{10.2} \pm \mathbf{0.8}$ | $1.4 \pm 0.4$ | $\mathbf{4.6} \pm \mathbf{0.9}$ |
| 500 | $5.8 \pm 1.5$ | $\mathbf{13.3} \pm \mathbf{0.6}$ | $4.1 \pm 0.9$ | $\mathbf{6.6} \pm \mathbf{0.3}$ |
| 1000 | $6.1 \pm 2.7$ | $\mathbf{13.3} \pm \mathbf{1.0}$ | $4.9 \pm 1.8$ | $\mathbf{7.9} \pm \mathbf{0.8}$ |

Table 14: Comparison of our method using Distribution Matching (**Ours w/ DM**) and Trajectory Matching (**Ours w/ TM**) on the Flickr30K dataset. The table shows retrieval performance (R@1) across different numbers of pairs. The results indicate that our method with trajectory matching consistently outperforms distribution matching, particularly in scenarios with smaller data budgets.

Looking forward, we hope our method could serve as a roadmap for future studies exploring more complex settings with new state-of-the-art (SOTA) methods. New SOTA dataset distillation methods can adopt low-rank adaptation matching to scale efficiently with large and complex models, and can incorporate bi-trajectory co-distillation to handle textual data more effectively. By doing so, these methods can extend their applicability to previously infeasible models for distillation, such as those involving ViTs, thus improving the scalability and efficiency of the distillation process. New approaches that distill from both text and image data can consider using methods similar to bi-trajectory matching with contrastive loss to learn the interactions and redundancies across multimodalities.

## G  Additional Visualizations

Here we include a number of visualizations of the data we distilled from the multimodal dataset (both Flickr30K Tab. G and Fig. 8, 9 and COCO Tab. G and Fig. 10, 11) for a more intuitive understanding of the distilled set. We provide 50 distilled image-text paired examples including their visualization before the distillation process. Unless otherwise stated, these experiments are conducted using 100 distilled pairs, with pretrained NFNet (Brock et al., 2021b) and BERT (Devlin et al., 2018) as backbones and the synthetic step is set to 8 during distillation. We provide visualization of distilled data using NFNet and CLIP in Tab. G and Fig. 12, 13 in the end.

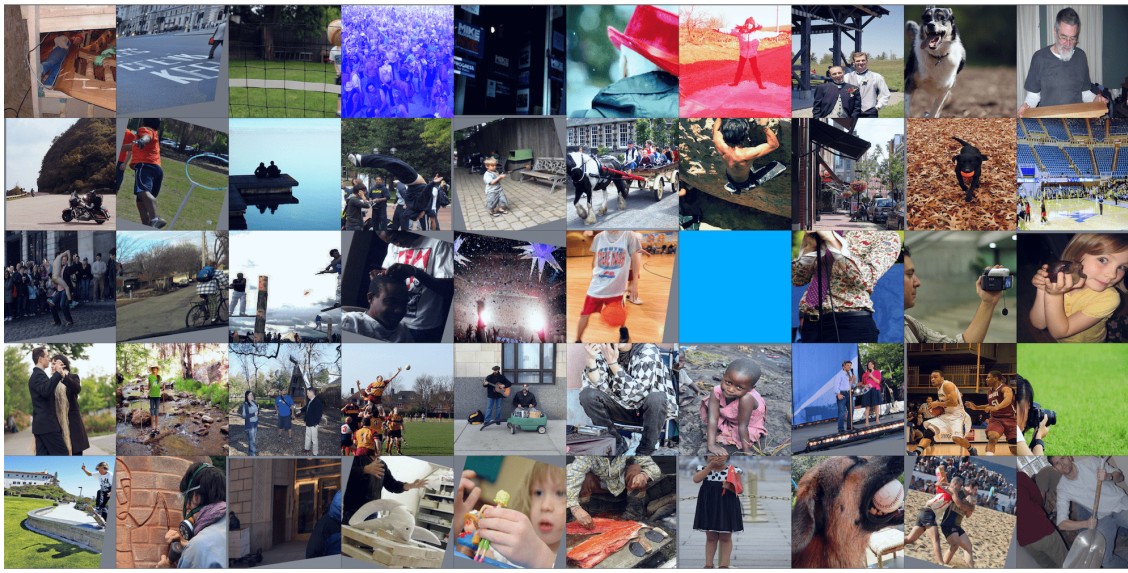

Figure 8: Flickr30K Initialized Images, iteration = 0.

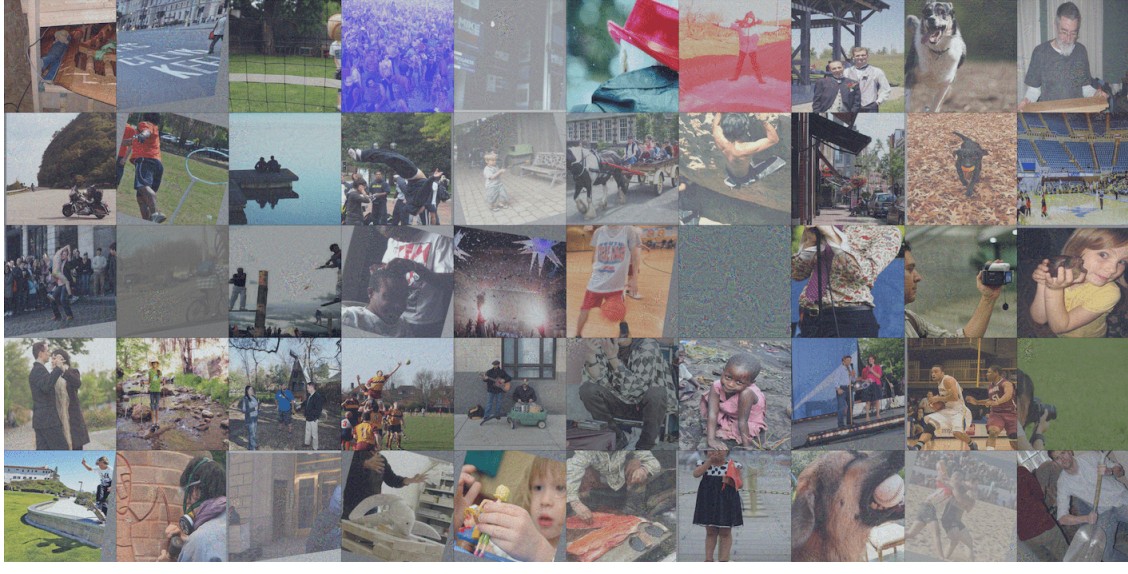

Figure 9: Flickr30K Distilled Images, iteration = 2000.

**Flickr30k Initialized Texts, iteration = 0.**

```
a construction worker stares down a flight of stairs being built
a man in a suit walking across a city street
a child hits a baseball into a net
a large crowd of people
an old man, behind him on glass is many political advertisements
an old man with white hair in a red hat
a young girl, on a road, playing on the ice
two men dressed up before a big event
a dog is running through a field with its tongue hanging out
an older man in a gray shirt with a white long-sleeve shirt under it holding up a small wooden cabinet
a motorcycle is parked along side a mountain road while another goes down the road
a man in an orange shirt and black shorts throws a ball through a hoop while another man watches
two people sit on the end of a dock
a man performs a back flip while preparing for an outdoor performance or competition
a little boy plays with a toy gun
a bunch of young children are riding on the back of a trolley while being carried by a black and white horse
a man climbing up on a rock ledge
a young man sits on a bench in a downtown setting
a black dog carries an orange ball, walking on the ground covered in leaves
basketball players practicing for their game
a man in just his underwear jumping on a man surrounded by a crowd of people
a man wearing lots of plaid riding a bike through the streets
men are trying to cut down trees
a black man getting a haircut
this was a big new years event the people were sing and dancing all night
a boy wearing a steve nash shirt dribbles a basketball on an indoor court
a series of men taking a break from riding their motorcycles
a brown-haired man in a patterned shirt and purple tie is singing into a microphone
a curly dark-haired man holds a small camcorder and films in a person in front of him
a young girl in a yellow shirt holds a rather large snail in her hands next to her cheek
two young men dancing in the street
a girl standing in a shallow creek, wearing stilts
3 people standing on a park talking to each other
a group of young men in colorful uniforms playing with a white ball
two guys are on the side of the street playing a guitar and drums
a mime applying his makeup
a child decorates a shoe with colorful sticks
a man and a woman are up on a stage with microphones in their hands
two basketball players on opposing teams, one in white, the other in red, are mid-game, running down the court, white-uniform
player with ball-in-hand
one young lady with black hair in a ponytail wearing a black bracelet and a white shirt, taking pictures with a black camera
that has a shoulder strap laying in
a boy on a skateboard is on a wall near the water and next to grass
a sculptor is carving a picture of a knight into a brick wall
a man in a blue coat is walking on the sidewalk
an old man wearing glasses is holding a stick
child playing with doll-like toy
an old man wearing a hooded sweatshirt is crouched over a fish that has been cut open
a young girl in a black dress is holding a red flag and covering a happy expression
a brown dog with a baseball in its mouth
man in white and red tackling man in green shirt for the ball
a man in a white t-shirt is holding a snow shovel
```

**Flickr30k Distilled Texts, iteration = 2000.**

construction workers repair walls of a subway
a ship in a harbor at night with a city skyline behind
baseball pitcher throwing a pitch
group of people sitting around a table for a meeting
a man points to something as he is talking to a woman wearing white pants, as they stand in front of a store
man in red sweater with a backwards hat
women wearing winter coats crossing the street next to parked cars and walking down street
the bridal party poses with the bride and groom, all wearing black except for the bride
an old lady, wearing a red hat, is standing on the sidewalk of a park
a man grilling hotdogs and sausages
a motocross bike kicks up dirt as it is being ridden around a bend in the circuit
nine women in blue and purple dresses and one man wearing a purple shirt and black pants, clap while a man dressed in black
dances
two young men and two boys are sitting down on a boat next to an anchor and watching the water
while playing soccer, a man in yellow starts to fall, while a man in white trips over him, stepping on his ankle in the
process
a little boy is walking on a fallen tree in the woods
a jockey and horse in the middle of other jockeys and horses during a race, in the middle of jumping over a hurdle
an extreme man snowboarding up side down a mountain
one man, in a blue jacket, is sitting in the rain under a green umbrella
the brown and white dog is running to catch something
boy takes a bath with diving mask and snorkel
angry looking businessman walking down sidewalk
a person on a bmx bike, leaping onto a bench
two people and a dog are in the snow
young shirtless boy sleeping on a couch with his hand on his chest
a person spins a sparkler around at night and sparks fly through the air and across the ground
a woman, wearing sunglasses, a red athletic top, and running shorts competes in a marathon
a ballet dancer wearing a blue tutu doing the splits, mid-leap
woman on street corner smiles and talks on her cellphone
policeman taping off an area by a group of firemen
a man with a beer and another man facing each other, talking
a woman and two children reading outside on a stone bench
man on motorcycle riding in dry field wearing a helmet and backpack
a man in a white shirt and black exercise shorts walks on a sidewalk, which is located behind a street under construction and
in front of a two garage house
a man is hitting a golf ball out of a sand trap, there are green grass all around him
some young adults are playing saxophones and clarinets outdoors
a young man sitting on a rock on the shore of a body of water looking contemplative
a young boy, covered in mud, plays on the beach
shirley manson poses in front of a microphone on stage while holding a large blue, red, and white flag behind her
two hockey players playing offense and defense
a group of friends, 3 boys and 2 girls, jump in the air holding hands for a photo
a boy skateboards and does a jump over another skateboard
ginger baby playing with a train setup made out of counterfeit lego
a group of choreographed rollerskaters dancing
little blond girl in her jacket sticking out her tongue while holding a red balloon
a blond little girl wrapped up in a pink care bears blanket
an oriental man, wearing a white shirt and apron is cooking
one boys sits on a giant mortar gun as another boy walks toward him
brown dog trying to bite a white ball with yellow, green and blue puppy toes
woman standing on the shore of a beach
2 males, one in a red shirt and one in a yellow shirt, cleaning a room

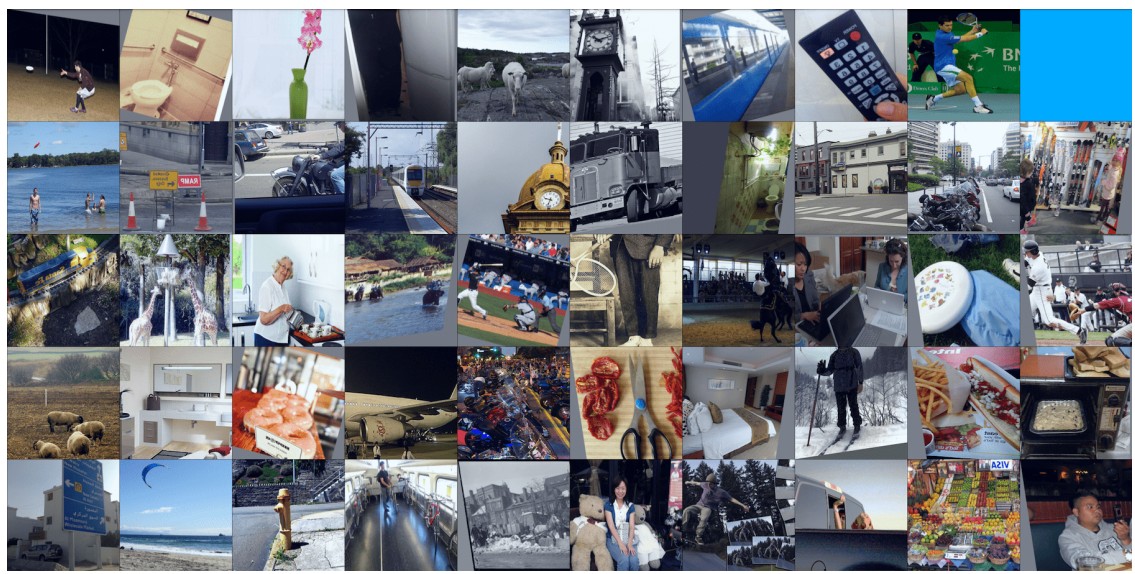

Figure 10: COCO Initialized Images, iteration = 0.

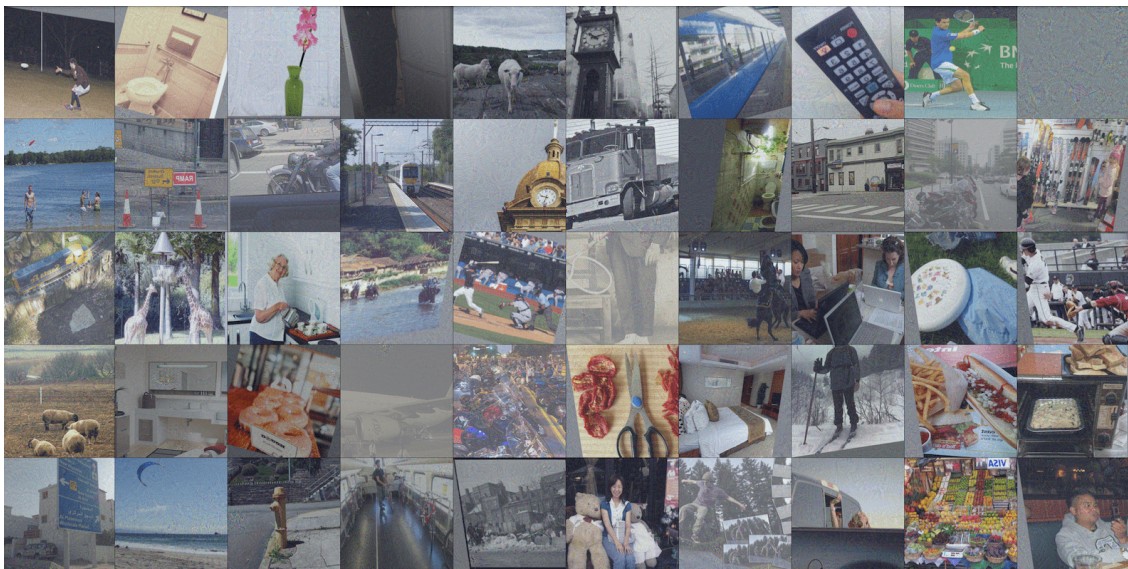

Figure 11: COCO Distilled Images, iteration = 2000.

**COCO Initialized Texts, iteration = 0.**

```
a photo taken at night of a young man playing frisbee
a bathroom toilet is surrounded with silver handrails
a pink flower is sticking out of a green vase
a woman poses next to a fridge
a small group of sheep on the coast
a black and white photo showing a large clock tower on a building
the people are waiting to be picked up
a hand holding a black television remote control
a man swinging a tennis racket at a tennis ball
a man holds a small animal to his face
three people are in the water as a frisbee is in the air
yellow and red street signs warning larger vehicles in a large city
a man riding on the back of a motorcycle
the train is traveling down the railroad tracks
a building with a large clock on it
a man standing on a wheel of a big tarck
a small bathroom with a toilet, sink and window
a florist shop and other buildings on and near a street corner
a line of motorcycles parked on a street
two young girls in a store looking at skis
a small toy model train on a track
a couple of giraffes are outside in the wild
a woman pouring coffee into cups on a counter
people riding on top of elephants across a river
a hitter swings at the baseball and misses
an old picture of a guy holding a tennis racquet
there are people watching three men on horseback
two people sitting at a table with laptops
there are many things laying on the ground
a batter swings hard at a low ball as the catcher reaches out his glove
five sheep stand around a large dirt field
a bathroom with a sink, mirrors and chair
glazed donut sitting on a wooden table top in a donut shop
a huge chinese aircraft is sitting at an airport with people unloading
a bunch of motorcycles are parked together outside
cutting board with scissors and dried food on it
a clean, decorated bedroom is pictured in this image
a man standing on skis at the top of a hill under high tension wires
a long hot dog and french fries are on a plate
a pan of dough is going into the dirty toaster oven
a road sign with both english and arabic directions
a kite being flown on the beach while people watch
a yellow fire hydrant is on the corner of an old sidewalk
a man with a bicycle and a helmet on his head in a subway car
a horse struggles to draw a loaded cart through piles of snow
a woman is sitting between two large teddy bears
a kid skateboarding while other kids stand and watch
there is a dog in the back of the truck
a large assortment of fruits lie on display in a market
he's taking a picture of his friends at the restaurant
```

**COCO Distilled Texts, iteration = 2000.**

```
dog flying in mid-air running after a frisbee
bath tub with metal shower head, vanity mirror, and small utilities compartment
a white vase with pink flowers and large green stems
this apartment has an kitchen with a refrigerator, stove, dishwasher, and cabinets
a ski resort with many people gathered around the outside lodge
grandfather clock hanging on wall next to a grandfather clock
the kitchen is brightly lit from the window
someone is playing with a nintendo wii controller
a woman swinging a tennis racket, while standing on a tennis court
a jet plane taking off into the air
a sign warning people to stop at red lights
man swinging baseball bat with ball in air and crowd watching
a man hitching a trailer with water sports equipment to a sports utility vehicle
four trains lined up at the train station
a large tower has a clock towards the top
people standing at a bus stop about to board a bus
a small bathroom with a sink a mirror
man admiring a motorcycle in parking lot, near a large building
a motorcyclist in a red and white suit riding a red and white motorcycle
a little boy playing tennis on a tennis court
a locomotive train resting on some tracks next to a building
two giraffes graze on some tall plant feeder
woman looking at camera while lying in bed
a herd of adult elephants with a baby elephant waling through a forest
baseball batter in a wide stance, waiting for a pitched ball
the cars were parked along the street near the traffic light
the travelers are getting around by horses
a cluttered desk with a laptop opened to flickr
the lady is sitting on the wood bench
a baseball player is preparing to swing a baseball bat
two lambs eating hay from ground of a field
some brown cabinets a black oven a tea kettle and a microwave
the reception ifs full of professional people
baseball batter ready to strike arriving ball and umpire waiting to catch if he misses
there lot of motorcycles park in a line with a white car, a red car and a van park not far from the motorcycles while there is
man riding on
a very clean room and a pair of scissors
the bedroom has a wood closet and bookcase near the bed
a line of skiers heading towards a cabin in the mountains
a plate topped with onions rings next to a hamburger and hot dog
the personal sized pizza has been topped with vegetables
a sign letting people know about the castle rising castle
girl and two males standing on a beach
there is a white fire dyrant on the corner street
a man skateboarding on street in front of a bus
vintage black and white photograph of two baseball players
raggedy ann doll sitting in a chair with a pooh bear
blonde haired boy doing a jump while riding a skate board
a dog driving a car down a street
there are bananas, apples and oranges in the bowl,
a stop light with the green light lit
```

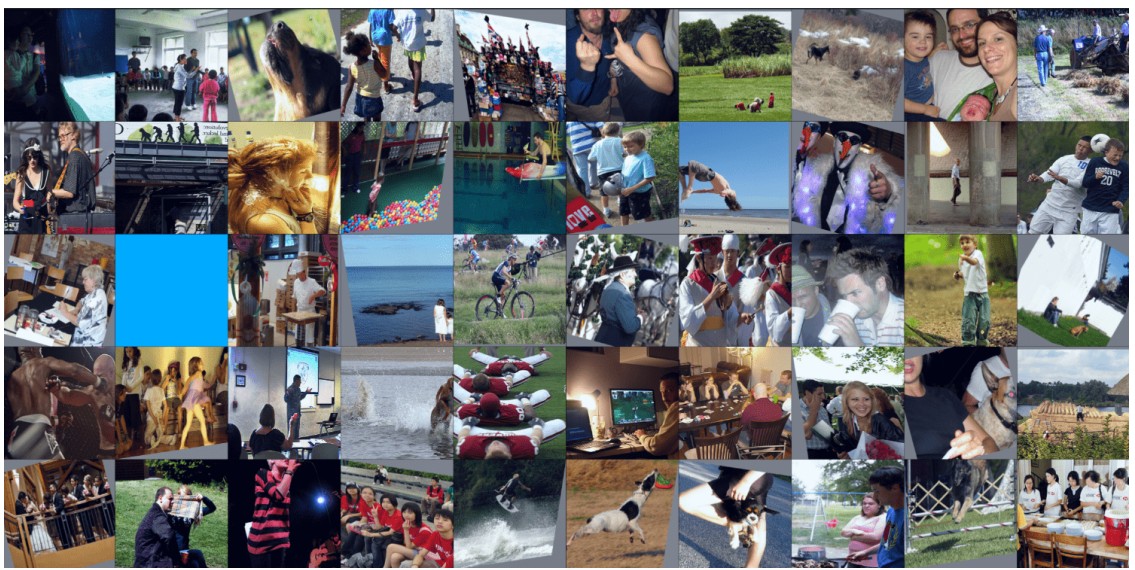

Figure 12: **CLIP**, Flickr30K Initialized Images, iteration = 0.

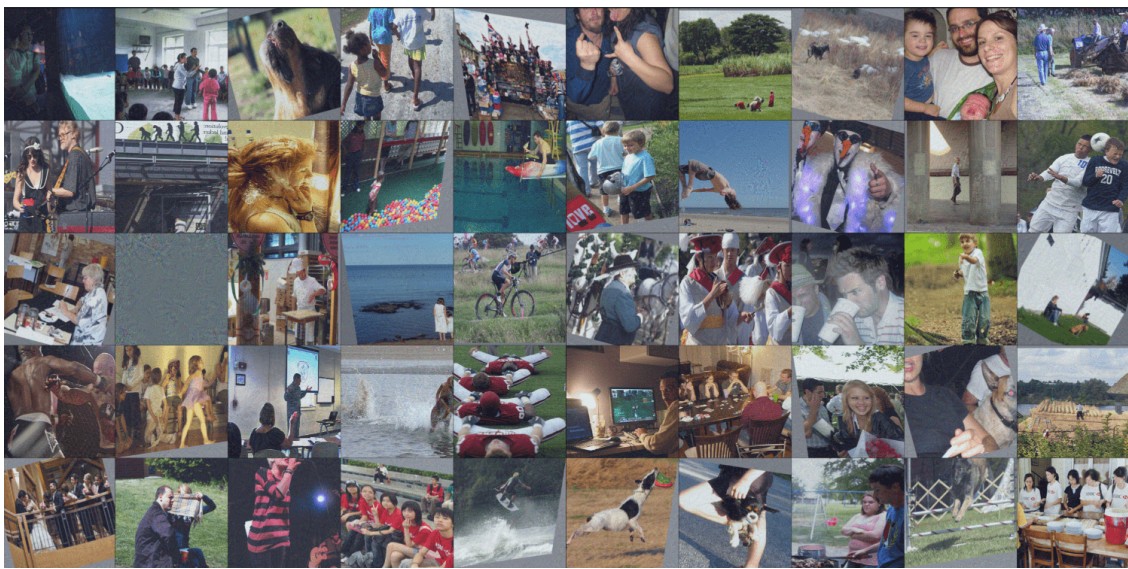

Figure 13: **CLIP**, Flickr30K Distilled Images, iteration = 2000.

**CLIP, Flickr30k Initialized Texts, iteration = 0.**

a woman holding a child is standing in front of a tank
a woman and man with a child in the center of a circle of children
a black and brown dog eyeing a fly
four kids next to a blue house are walking down a street
tourists look at merchandise on a street vendors display in england riffling through cards and maps
two people wearing blue clothing are making hand gestures next to one another
four workers in a field harvesting
a dog approachs a small creature in a barren, snowy field
a woman holding a baby and a man in glasses holding a small boy smile at the camera for their family photo
four men are harvesting a crop on a farm
two musicians on stage in front of a microphone stand
a man in a suit is walking under a metal bridge
girl getting her hair dyed
a child is in a ball pit while three adults watch her
a man is sitting in a small kayak on a diving board
a boy in a blue shirt holds a toy helmet in his hands while standing on a path in a park
a boy doing a flip in the air at the beach
two men are dressed up as snowmen
a man walking underneath a bridge glances at the camera
two soccer players are about to butt heads getting the ball
a woman is at an art studio, painting a mural from her art supplies
a boy wearing a black wetsuit stands on a crowded beach
a chef prepares a table with food behind it
two blond girls in white dresses, one much smaller than the other, stand on the bank of a large body of water
a man on a black mountain bike rides through a course filled with other bikers
a man in a hat stands with decorated horses around him
an asian marching band with uniformed members in beige, yellow, and red play in the street
two men with angry faces drink out of white cups
young boy plays with leaves in a green wooded area
a person siting against a wall with a dog
one fighter delivers a hard blow to the face of another fighter
a young group of children sitting in a row against the wall
a teacher stands in front of a projector and a student at the front of the class has her hand raised
a dog is running in a large body of water causing it to splash
football players are stretching together
man playing video game instead of working
a group of people at dining table playing a card game
young woman celebrating her graduation
a dog looks on as a woman eats
a man wearing a flannel shirt and black pants is working on a new reed roof on top of a house
newly married couple having their first kiss
a woman plays hide-and-go-seek with a check scarf as she sits with a man in a dark colored jacket
a woman with short blond-hair smiling and singing into a microphone while a man in a striped shirt, further back, plays an acoustic guitar
a group of asian teenagers wearing red t-shirts sit on steps
a man gets lots of air time as he wakeboards
a black and white dog is running through the field to catch something in its mouth
a person with a small dog caught in her legs
a young man tends chicken wings on a barbecue while a young woman in a pink shirt watches
black and brown dog jumping over hurdle with white supports
a group of women wearing shirts that say, hsbc is standing by a table with food on it

CLIP, Flickr30k Distilled Texts, iteration = 2000.

a young asian girl petting an animal of some sort and the animal is laying down enjoying it
a group of men in ethnic dress are dancing
brown and tan dog, mouth open with tongue hanging out, running in the grass
an older black woman wearing a colorful print dress stares directly at the camera
woman in red shirt shopping in a outdoor market
a woman in a blue shirt with no bra
a man holding a bag walking down a long staircase
a man in black walking down a street
a woman holds a baby in a blue jumper
a small car in an open field
a group of male musicians are playing instruments including guitar and drums
a man is standing inside a subway train with his mouth wide open
a barber shaving someone's head
a group of people are shopping in what looks to be a christmas store filled with colorful toys
a man in high rubber boots and a plaid shirt is pushing a broom over the mossy blacktop
three males with cameras near each other, two sitting and the third standing, in what might be a park during a sunny day
a girl jumping up in the air with her hands above her head
two people, one of whom is in a santa costume, pose wearing funny glasses
a group of people walking through an alley along a cobblestone street, between two buildings
a soccer player in a green jersey kicks a blue and yellow ball
a man painting over graffiti
two dogs running near a river while one dogs in swimming in it
a man in a black hat looks surprised at the deli counter
a woman sits in a chair on the beach, backdropped by the ocean
a young man popping a wheelie on his bicycle while riding down a country road
a person attempts to rope a black cow while riding a horse
men dressed in red and white playing musical instruments
two men drink beer out of tall drinking glasses
a little boy is eating on a sidewalk
a man is standing inside a doorway that is in a wall painted with a mural of a woman
a man wearing a hat has his eyes closed, as another man in a red shirt is licking his face
a family of 3 sits and poses on a couch together
a young boy in a sports uniform stands in front of a group of children
a little boy plays outdoors in water spurting up from an inground fountain
two men in red pants do acrobatics with a ladder
a young caucasian man sits at a desk using a laptop computer
a group of people is sharing a meal at a large table at a restaurant
a group of people protest with one holding up a cardboard sign
a group of people and their dogs at a dog show
a man with a plaid shirt is working on some wood in his workshop
the fellow in the black suit at a formal occasion has a salmon rose in his lapel
a man and a woman walking across a field of grass
a woman singer holding a microphone singing at a concert
young asian female sitting in a pose on a stone wallas she is being photographed
the man is standing by a creek in blue flannel shorts
a black and white dog leaps to catch a frisbee in a field
a man is feeding two exotic birds
an asian chef is in the foreground, working over a steaming grill while a younger man is behind him
a young man is skateboarding down the railing of some stairs
a female chef examines a piece of bread while showing it to the camera

