# OpenReview forum: "Vision-Language Dataset Distillation"
_TMLR — Accepted by TMLR_

### Review · Reviewer_CQga · 2024-05-23

**Summary Of Contributions:**

The paper introduces a novel approach for dataset distillation specifically tailored for vision-language models, a domain that has not been previously explored. The main contributions are as follows:

- Building upon the state-of-the-art trajectory matching method, the authors adapt this technique to vision-language datasets to co-distill both images and text.

- Unlike previous dataset distillation efforts that have concentrated on image classification in a continuous space with fewer parameters, the authors propose using Low-Rank Adaptation (LoRA) to address the challenge of managing the large number of parameters in the new domain.

- The paper demonstrates the effectiveness of the proposed method by comparing it to three core-set selection methods.

**Audience:**

Yes

**Claims And Evidence:**

Yes

**Requested Changes:**

The paper could be improved by discussing how the proposed method can be adapted by new state-of-the-art methods that go beyond trajectory matching. [not critical for my recommendation]

**Strengths And Weaknesses:**

## Strength
- The paper is well written and easy to follow
- The method is quite novel such as adopting LoRA as methods like MTT cannot easily scale to large number of parameters due to heavy memory usage.
- The experiment results prove that the proposed method is very effective in distilling image together with text.
- This is the first work to tackle the problem of dataset distillation for vision-language models to the best of my knowledge.

## Weakness
- Since the method is based off MTT, can the proposed method easily extend to using other DD method as the backbone?
- Have the authors explored other methods for distilling text other than the input embedding space?

---

> ### Author Response · Authors · 2024-06-04
>
> Thank you for taking the time to review our work and to suggest improvements!
>
> - Yes, we can in fact extend our framework to other DD methods beyond MTT. Here we further provide experimental results of a distribution matching baseline adapted to the vision-language setting. To use distribution matching for vision-language dataset distillation, concretely, we minimize the maximum mean discrepancy (mmd) between two distributions by sampling NFNet with different initializations and pretrained BERT. Similar to the distribution matching setting for image classification, we update the distilled data via mmd for vision and language modalities to match the original data distribution in a family of embedding spaces. Here is the result table for Our method w/ DM and comparison with Our method w/ TM on Flickr30K (R@1).
>
> |          |            |     TR     |          |      IR    |          |
> |-----------|---------|------------|----------|------------|----------|
> |   Dataset   |   # pairs  | Our method w/ DM | Our method w/ TM    | Our method w/ DM | Our method w/ TM    |
> | Flickr30K | 100     | 3.2 ± 1.8  | 9.9 ± 0.3 | 1.4 ± 0.7  | 4.7 ± 0.2 |
> |           | 200     | 3.3 ± 1.3  | 10.2 ± 0.8| 1.4 ± 0.4  | 4.6 ± 0.9 |
> |           | 500     | 5.8 ± 1.5  | 13.3 ± 0.6| 4.1 ± 0.9  | 6.6 ± 0.3 |
> |           | 1000    | 6.1 ± 2.7  | 13.3 ± 1.0| 4.9 ± 1.8  | 7.9 ± 0.8 |
>
> - There is one concurrent work [1] that focuses on distilling a text dataset into a sentence-level synthetic dataset instead of the word embeddings. They bypass the discreteness of text by introducing a language model as a surrogate optimization target and backpropagating the distillation loss to the model. However, this method introduces significant computational overhead due to the size and complexity of the GPT-2 model used, which could be impractical for many applications. We intentionally opted for a method where the text embedding is chosen a priori and remains fixed throughout the distillation process. To our knowledge, there hasn't been a single paper successful in performing text distillation directly in the discrete word space (even [1] bypass the non-differentiable generated text), which would represent the most transferable form of distilled text. This limitation led us to adopt a strategy centered around a specific embedding, focusing on distillation within this predefined space. However, our method is agnostic of the particular choice of embedding, and in the future if a method were to be developed which can distill from e.g., the word space directly, or from a transferable text embedding, it (likely) could be seamlessly integrated into our framework.
> [1] Maekawa, Aru, et al. "DiLM: Distilling Dataset into Language Model for Text-level Dataset Distillation." arXiv preprint arXiv:2404.00264 (2024).
>
> Thanks for the requested changes! We have highlighted the changes in blue in the updated version of the paper (Appendix Sec. F).

---

> > ### Comment · Reviewer_CQga · 2024-07-08
> > **Thanks for the responses**
> >
> > Thank the authors for the responses to my questions. My concerns have been resolved. I think this is a valuable paper that's worth sharing with the community for developing future vision-language dataset distillation/condensation methods. Based on the questions from other reviewers and the corresponding responses from the authors such as extending the language distillation part to other methods, it can be seen that the proposed method is indeed a strong baseline method in distilling vision and language datasets (with room for improvement).
> >
> > At the same time, I agree with reviewer aeRf and zbg5 that the paper presentation could be greatly improved by addressing the requested changes from the reviewers (the updated version has showed great progress). Please make sure to address all requested changes and integrate the extra experiment results into the next version of your paper. Conditioning on that, I recommend acceptance of the paper.

---

> > > ### Author Response · Authors · 2024-07-09
> > > **Thank you!**
> > >
> > > Thank you for your positive feedback on our work! We are pleased to hear that our responses addressed your concerns and are committed to refining our paper based on your insights. Thank you once again for your thorough review and support!

---

### Review · Reviewer_zbg5 · 2024-05-25

**Summary Of Contributions:**

This paper extends the research on classification dataset distillation to vision-language dataset distillation. The authors adapts the popular trajectory matching method for vision-language model optimization by introducing the bi-trajectory matching. Experiments show that the proposed method works much better than the coreset selection baselines. Visualization of the learned synthetic data is given for analysis.

**Audience:**

Yes

**Broader Impact Concerns:**

No.

**Claims And Evidence:**

Yes

**Requested Changes:**

No.

**Strengths And Weaknesses:**

Strengths:

1. The research problem of vision-language dataset distillation is interesting and novel.
2. The paper is well-written and well-motivated. The proposed method is reasonable and easy to re-produce.
3. Experimental results prove that the new method works much better than the baselines.

Weaknesses：

1. The performance of the model trained on distilled dataset is poor, which prevents the utility.
2. As shown in Figure 3, the distilled image/text pairs are somewhat unexplainable. For example, the unexist concepts in texts and the artifacts in images.

---

> ### Author Response · Authors · 2024-06-04
>
> We sincerely appreciate your feedback. Below we address the two noted weaknesses:
>
>
> - In the appendix Sec. C Tab. 8, we show that with only 10% of the full Flickr30K dataset, models trained on the distilled data show remarkable performance, closely approaching the upper bound results. We agree that the performance of the model trained on a very small scale distilled dataset leaves room for improvement. However, the goal of this work was to explore a novel dataset distillation approach for multimodal learning and we’ve shown significant performance improvements compared to baseline approaches.
>
> - Regarding the unexplainable nature of the distilled image/text pairs, the goal of dataset distillation is to compress the information from the original dataset into a smaller, synthetic dataset while preserving the most critical information. As a result, the distilled data is not meant to be directly interpretable or explainable in the same way as the original data. The data is optimized for performance rather than generating realistic or easily understandable examples for humans.

---

> > ### Comment · Reviewer_zbg5 · 2024-06-28
> >
> > Thanks for the response. Though there are few minor problems, the overall quality of work is good. I tend to accept it.

---

> > > ### Author Response · Authors · 2024-06-28
> > > **Thank you!**
> > >
> > > We are delighted to hear this and we really appreciate your valuable insights and the time you have invested in reviewing our work. Thank you for your positive feedback!

---

### Review · Reviewer_aeRf · 2024-06-12

**Summary Of Contributions:**

The submission gives an overview of a method for distilling a dataset for vision-language pretraining. The method is evaluated on a number of vision-language tasks, and the results are compared with several core-set methods. The method is shown to outperform the core-set methods by a substantial margin.

**Audience:**

Yes

**Claims And Evidence:**

No

**Requested Changes:**

* The authors should make it clear how they go from the distilled token embeddings to the discrete tokens.
* The authors should compare their method with other dataset distillation methods.
* The authors should improve the presentation of the evaluation, so that it is easier to understand the results. In particular, I would recommend that Table 1 is removed and replaced by plots summarising the results in Table 6.
* Please add performance of the models trained on the whole dataset to each of the tables, so that it is easier to understand the practical significance results.
* The writing needs some work to tone down the overclaiming and to generally better reflect the contributions actually being made in this paper.

**Strengths And Weaknesses:**

* The paper does a good job of introducing the new setting that is considered, and the method is described in reasonably good detail. One thing that is unclear is how one goes from the distilled token embeddings to the discrete tokens. This should be made more clear
* Experimental evaluation looks at several different aspects, including comparison with prior methods, an ablation, qualitative results, and generalisation of the distilled datasets to new architectures. In that sense, it is quite a good evaluation. Moreover, the method is shown to outperform core-set methods by a substantial margin.
* I like that the submission also considers the case of dataset distillation in the case of model adaptation---i.e., when training Low Rank Adaptations of the model.
* However, there are some things missing from the evaluation: there should be a comparison with other dataset distillation methods. The authors claim that this is not possible because text is discrete and therefore cannot be optimised via gradient-based optimisation. However, the method used in this paper optimises token embeddings, not discrete tokens, so it should be possible to compare with other methods by using the same token embedding to discrete token transform employed by the current method.
* The presentation of the evaluation could be improved. It is currently quite hard to understand how meaningful the results are, and the tables in the main paper do not provide the full picture.
* The main issue with the submission is overclaiming of novelty; the method is from a previous paper, and the only new thing is the evaluation in the vision-language setting. This is fine, but the way the paper is written is not in keeping with the contributions being made.

---

> ### Author Response · Authors · 2024-06-24
>
> We sincerely appreciate your feedback. Below we address the weaknesses:
> - **Distilled token embeddings to discrete tokens.**
> We actually mentioned how to transform from distilled token embeddings to discrete tokens in the caption of Fig.3 and Sec. 4.2, qualitative results. Note that the discrete tokens are not the final distilled data saved in the distilled dataset, and only for visualization purposes.
> - **Comparison with other dataset distillation methods.**
> We want to clarify that we never claimed “comparing with other dataset distillation method is impossible due to text is discrete”, the original sentence in the paper is: “Lastly, unlike continuous data, text is inherently non-differentiable, making direct gradient-based optimization impossible on discrete text tokens.”
> As we are the first vision language dataset distillation method, here we further conduct experiments to compare with a distribution-matching baseline adapted to the vision-language setting. To use distribution matching for vision-language dataset distillation, concretely, we minimize the maximum mean discrepancy (mmd) between two distributions by sampling NFNet with different initializations and pretrained BERT. Similar to the distribution matching setting for image classification, we update the distilled data via mmd for vision and language modalities to match the original data distribution in a family of embedding spaces. Here is the result table for Our method w/ DM and comparison with Our method w/ TM on Flickr30K (R@1).
>
> |          |            |     TR     |          |      IR    |          |
> |-----------|---------|------------|----------|------------|----------|
> |   Dataset   |   # pairs  | Our method w/ DM | Our method w/ TM    | Our method w/ DM | Our method w/ TM    |
> |-----------|---------|------------|----------|------------|----------|
> | Flickr30K | 100     | 3.2 ± 1.8  | 9.9 ± 0.3 | 1.4 ± 0.7  | 4.7 ± 0.2 |
> |           | 200     | 3.3 ± 1.3  | 10.2 ± 0.8| 1.4 ± 0.4  | 4.6 ± 0.9 |
> |           | 500     | 5.8 ± 1.5  | 13.3 ± 0.6| 4.1 ± 0.9  | 6.6 ± 0.3 |
> |           | 1000    | 6.1 ± 2.7  | 13.3 ± 1.0| 4.9 ± 1.8  | 7.9 ± 0.8 |
>
> We include these results in the updated version of our paper (Appendix Sec. F).
>
> - **Presentation.**
> In the previous version of our paper, we included the full performance data in the main table, but found it distracted from the main message which is the fact that our method outperforms baseline methods by a large margin in the data-limited settings. Instead, we provide the full performance data, along with results using 10% of the distilled data, in the appendix Sec. C for clarity. Including all this information in the main table can be confusing. We have added the following caption to Tab. 1: 'In the very small budget regime, retrieval accuracy is much lower compared to the full data performance. As the budget grows, performance reaches full performance as shown in Tab. 8 in the appendix Sec. C.' We hope this clarifies our presentation approach and look forward to any further suggestions.
> - **Overclaiming of novelty.**
> We greatly appreciate your feedback and would be grateful for specific suggestions on which parts of our paper overclaim novelty. We are very concerned about this comment and committed to making sure we did not overclaim and want to address any overstatements promptly. We have carefully reviewed our paper again, and we actually only used “novel” once in the entire paper: Fig.1 caption, “In contrast, we set out to distill vision-language datasets with no discrete classes; we do so via a novel method which jointly distills images and texts.”. Could you please point us to the sections/sentences where we may have overclaimed so we can make the necessary revisions?
>
> We have highlighted the changes in blue in the updated version of the paper.

---

### Decision · Action_Editor_tadn · 2024-07-18

**Recommendation:** Accept as is

**Comment:**

The reviewers found various positive elements in the article, such as the presentation (CQga, zbg5), the addressed problem (aeRf, CQga, zbg5), the experimental results (aeRf, CQga, zbg5), and the idea behind the proposed approach (CQga, zbg5).

At the same time, they raised some concerns about the low overall performance (zbg5), lack of explainability (zbg5), the generality of the framework (CQga), novelty claims on the method (aeRf), and lack of clarity on some aspects of the evaluation and embeddings discretization (aeRf).

The authors' reply fully addressed the concerns of CQga and zbg5, who recommended full acceptance. While aeRf also leans to accept the manuscript, they raise some points on the presentation (i.e, tables vs plots) and on overclaiming novelty on the method (i.e., trajectory matching). The AE finds the latter concerns minor as i) the presentation quality of the results might be subjective and the AE finds no critical issues in the current version; ii) the article stresses in multiple points that the proposed method is based on trajectory matching, providing proper references and descriptions (i.e., Section 3.3).

Considering the revised article and the reviewers' feedback, the AE recommends the acceptance of the work.

**Audience:**

Dataset distillation is a popular line of work aiming to compress the size of training datasets. This article extends this direction to vision-language models: these results might be interesting not only for researchers in dataset condensation but also on multimodal learning, where the findings can be applied.

**Claims And Evidence:**

The work studies the problem of dataset distillation, i.e., reducing the size of a large dataset into a smaller, synthetic version that allows models to achieve the same results as the original dataset when trained on it. In particular, this paper is the first to study this problem in the context of vision-language datasets. The article presents an approach based on trajectory matching and tests several distillation methods on this new problem, showing the efficacy of the proposed approach.

All reviewers agree that the core claims, i.e., being the first to study this problem and the efficacy of the proposed method, are valid, with the latter confirmed by the experiments provided in the original (or revised version of the manuscript).